# CCR1 mediates Müller cell activation and photoreceptor cell death in macular and retinal degeneration

Sarah Elbaz-Hayoun, Batya Rinsky, Shira Hagbi-Levi, Michelle Grunin, Itay Chowers*

Department of Ophthalmology, Hadassah Medical Center, Faculty of Medicine, The Hebrew University of Jerusalem, Jerusalem, Israel

**Abstract** Mononuclear cells are involved in the pathogenesis of retinal diseases, including age-related macular degeneration (AMD). Here, we examined the mechanisms that underlie macrophage-driven retinal cell death. Monocytes were extracted from patients with AMD and differentiated into macrophages (hMdɸs), which were characterized based on proteomics, gene expression, and ex vivo and in vivo properties. Using bioinformatics, we identified the signaling pathway involved in macrophage-driven retinal cell death, and we assessed the therapeutic potential of targeting this pathway. We found that M2a hMdɸs were associated with retinal cell death in retinal explants and following adoptive transfer in a photic injury model. Moreover, M2a hMdɸs express several CCRI (C-C chemokine receptor type 1) ligands. Importantly, CCR1 was upregulated in Müller cells in models of retinal injury and aging, and CCR1 expression was correlated with retinal damage. Lastly, inhibiting CCR1 reduced photic-induced retinal damage, photoreceptor cell apoptosis, and retinal inflammation. These data suggest that hMdɸs, CCR1, and Müller cells work together to drive retinal and macular degeneration, suggesting that CCR1 may serve as a target for treating these sight-threatening conditions.

*For correspondence:
chowerslab@gmail.com

Competing interest: The authors declare that no competing interests exist.

## Editor's evaluation

Immune cell invasion, gliosis, and photoreceptor cell death are observed in multiple retinal diseases. This important study identifies cells and signaling pathways that connect these three processes during retinal degeneration. The authors provide convincing experimental evidence linking macrophages to the activation of retinal Müller glial cells and photoreceptor death. These results are significant as they identify cell types and potential targets linking immune cells to retinal cell changes, ultimately resulting in photoreceptor cell death.

## Introduction

Age-related macular degeneration (AMD) is the leading cause of irreversible blindness among the elderly in the Western world (*Friedman et al., 2004*; *Resnikoff et al., 2004*; *Singer et al., 2012*). Atrophic AMD (aAMD, also known as 'dry' AMD) is characterized by the progressive loss of retinal pigment epithelial (RPE) cells and photoreceptor cells, which can coalesce and cause geographic atrophy in the macular region. In contrast, neovascular AMD (nvAMD, also known as 'wet' AMD) is characterized by choroidal neovascularization (CNV) (*Leeuwen et al., 2003*; *Joachim et al., 2015*). Although anti-VEGF (vascular endothelial growth factor) compounds are used to treat nvAMD by reducing leakage and the progression of CNV (*Rosenfeld et al., 2006*), treatments that slow the progression of atrophy and vision loss in aAMD are lacking and therefore urgently needed.

AMD is caused by multiple factors, including senescence, genetic factors, environmental factors, and impaired immune system function (*Bowes Rickman et al., 2013*). Moreover, inflammation is a major factor involved in the pathogenesis of AMD (*Augustin and Kirchhof, 2009*). Polymorphisms in several genes encoding complement factor proteins such as complement factor H (*Smailhodzic et al., 2012*), C3 (*Yates et al., 2007*), and C5 *Baas et al., 2010* have been associated with both forms of AMD, suggesting an underlying dysregulation of the complement cascade. Increased levels of C-reactive protein and complement activation have also been measured in the plasma of patients with AMD (*Seddon et al., 2004*), and histological analyses of AMD eyes revealed the presence of macrophages in the vicinity of the atrophic lesion (*Cao et al., 2011*; *Lad et al., 2015*; *Buschini et al., 2011*), as well as in drusen (*Penfold et al., 2001*) and the subretinal space (*Sennlaub et al., 2013*). Moreover, a non-resolving immune response involving the sustained recruitment of immune cells has been shown to contribute to the development of various neurodegenerative diseases, including AMD (*Nathan and Ding, 2010*; *Apte et al., 2006*). Thus, determining the precise role that immune cells play in the onset and progression of AMD is a necessary step toward developing new therapeutic approaches.

We previously showed that peripheral blood mononuclear cells (PBMCs) obtained from patients with AMD have a pro-inflammatory gene expression profile (*Lederman et al., 2010*). In addition, other groups have shown that blocking monocyte recruitment reduces the degree of retinal injury in various animal models (*Sennlaub et al., 2013*; *Rutar et al., 2012*; *Yang et al., 2011*; *Raoul et al., 2010*; *Nakamura et al., 2015*; *Wang et al., 2019*). Moreover, modulating macrophage activation has been shown to alter photoreceptor cell survival both in Ccl2/Cx3cr1 double-knockout mice (*Shen et al., 2011*) and in a model of photic injury (*Ni et al., 2008*).

Upon infiltration, monocytes can differentiate into a variety of macrophage phenotypes. Although these phenotypes can have high plasticity and heterogeneity, two primary subtypes of macrophages—namely, M1 and M2 macrophages—represent opposite extremes of the polarization spectrum (*Mantovani et al., 2002*; *Martinez et al., 2008*). M1 macrophages are generally considered to be pro-inflammatory (*Mantovani et al., 2004*; *Sica et al., 2014*), while M2 macrophages are generally associated with tissue repair and remodeling (*Martinez and Gordon, 2014*; *Mills, 2012*). Interestingly, however, studies suggest that this functional dichotomy between M1 and M2 macrophages may not accurately reflect the plasticity of macrophage activity (*Hu et al., 2014*). In addition, other studies suggest that a variety of factors such as age (*Zhao et al., 2013*; *Sebastián et al., 2009*; *Kelly et al., 2007*; *Sene et al., 2013*), genetic background (*Tuo et al., 2012*), the tissue/organ involved, and the presence of underlying disease (*Lewis and Pollard, 2006*) can differentially affect both the function and response of macrophages. In the context of AMD, we previously showed that human monocyte-derived macrophages (hMdΦs) obtained from patients with AMD have a stronger pro-angiogenic effect than hMdΦs obtained from age-matched controls (*Hagbi-Levi et al., 2017*).

Specific chemokine receptors have also been implicated in AMD. For example, monocytes expressing CCR2 (C-C chemokine receptor type 2) have been found in the vicinity of geographic atrophy lesions, suggesting a pathogenic role (*Sennlaub et al., 2013*). However, which macrophage subtype is associated with cell death and/or the beneficial effects of drusen degradation in preventing RPE cell loss is currently unknown. Interestingly, CCR3 has been implicated in angiogenesis (*Sharma et al., 2013*), and CX3CR1 (C-X3-C motif chemokine receptor 1) has been implicated in the activation of microglial cells (*Combadière et al., 2007*).

CCR1, a G protein-coupled receptor, has been shown to play an essential role in recruiting leukocytes during inflammation (*Tsou et al., 1998*), and this receptor is expressed in a variety of immune cell types, including dendritic cells, neutrophils, T cells, and monocytes/macrophages (*Horuk, 2001*; *Gao et al., 1997*). We previously showed that monocytes obtained from a patient with AMD had increased expression of both CCR1 and CCR2 compared to monocytes obtained from an age-matched control (*Grunin et al., 2012*). Although CCR1 has been implicated in renal ischemia (*Furuichi et al., 2008*), rheumatoid arthritis (*Santella et al., 2014*; *Nanki, 2016*), endometriosis (*Xu et al., 2014*; *Trummer et al., 2017*; *Yang et al., 2013*), and multiple sclerosis, its role in AMD remains unknown.

Here, we investigated the signaling pathway involved in macrophage-mediated photoreceptor cell death, and we examined whether targeting macrophage signaling may serve as a potential therapeutic strategy for AMD.

## Results

### M2a hMdϕs increase photic-induced photoreceptor degeneration

We first compared the effect of monocytes from AMD and healthy individual on photoreceptor cells subjected to neurodegenerative conditions. To induce neurodegeneration, mice were exposed to bright light (photic retinal injury; *Figure 1—figure supplement 1A*). Light exposure caused photoreceptor cell death around the optic nerve head (ONH) (*Figure 1—figure supplement 1B, C*), while cells in the periphery were mostly spared (*Figure 1—figure supplement 1D, E*). Immediately after photic injury, the mice received an intravitreal injection of human monocytes derived from patients with AMD in one eye; the fellow eye was injected with human monocytes derived from unaffected age-matched controls. Eyes injected with monocytes from unaffected controls showed reduced electroretinography (ERG) b-wave amplitudes at various light stimuli intensities compared with eyes injected with monocytes from AMD patients (*Figure 1—figure supplement 1H*). This finding is supported by previous studies showing diminished functionality and plasticity of monocyte/macrophages from unhealthy patients, and particularly AMD, compared with controls (*Gu et al., 2021*). For example, an altered M1/M2 polarization balance (*Costantini et al., 2018*) and change in gene expression profile (*Grunin et al., 2012*) were reported in monocyte/macrophages.

Thus, the following experiments were focused on monocytes/macrophages derived from AMD patients to obtain a comprehensive understanding of the role of these cells in AMD progression. We examined whether monocytes obtained from AMD patients are neuroprotective or detrimental to photoreceptor cell survival in the photic injury model. Immediately after photic retina injury in mice, human monocytes from AMD patients were delivered via intravitreal injection in one eye; the other eye was injected with vehicle (phosphate-buffered saline [PBS]) as a control (*Figure 1—figure supplement 1A*). Compared with the control eyes, eyes injected with monocytes extracted from AMD patients showed reduced ERG b-wave amplitudes at various light intensities (*Figure 1—figure supplement 1I*), and increased photoreceptor cell loss in the dorso-central retina at distance ranging from −300 to −1200 μm from the ONH (*Figure 1—figure supplement 1J*).

Once recruited to the site of inflammation, monocytes can differentiate into variety of macrophage subtypes; we therefore evaluated for correlation between macrophage polarization and the neurotoxicity observed in the photic-injured retina. To that end, monocytes obtained from patients with AMD were polarized into M0, M1, M2a, and M2c macrophages by stimulation with M-CSF, LPS+IFNγ, IL-4+IL-13, or IL-10, respectively. Polarization of M0 macrophages into the appropriate activated hMdϕs was confirmed using qPCR to measure *CXCL10* (a marker of M1 hMdϕs) (*Yuan et al., 2015*; *Figure 1A*), *CCL17* (a marker of M2a hMdϕs) (*Hagbi-Levi et al., 2017*; *Mantovani et al., 2002*; *Figure 1B*), and *CD163* (a marker of M2c hMdϕs) (*Yuan et al., 2015*; *Figure 1C*). In addition, each hMdϕ subtype displayed a distinct morphology (*Figure 1D–G*). Polarization into M1 and M2a hMdϕs was also validated by immunohistochemistry for CD80 and CD206, respectively (*Figure 1—figure supplement 2A, B*).

When we tested the effects of polarized hMdϕs in our photic retinal injury model, we found that M2a hMdϕs suppressed ERG b-wave amplitude at various light intensities (*Figure 1J and N*) and accelerated photoreceptor cell loss from −300 μm to −1200 μm from the ONH (*Figure 1Q and U*) compared to control eyes. In contrast, M1 hMdϕs, which have been reported as pro-inflammatory cells in other organs (*Cruz-Guilloty et al., 2013*), had no effect on ERG b-wave amplitude or outer nuclear layer (ONL) thinning compared to control eyes (*Figure 1I and P*). Similarly, neither M2c hMdϕs nor M0 hMdϕs affected photoreceptor cell death (*Figure 1H, K, O, R*).

Immunohistochemistry for VEGF and CD206 (mannose), both accepted as M2a markers (*Kadomoto et al., 2021*; *Brüne et al., 2015*), demonstrated that monocytes from AMD patients polarized into M2a phenotypes following delivery to the mice eye in the photic retina injury model (*Figure 1—figure supplement 1K–S*). This finding may explain why monocytes from healthy individuals showed higher neurotoxicity compared with monocytes from AMD patients. Our previous research (*Hagbi-Levi et al., 2017*) indicated that M2a hMdϕs from healthy individual were associated with increased expression and secretion of pro-inflammatory proteins such as IL-6, CCL2, SDF-1, VEGF, and PDGF-α, compared with M2a hMdϕs from AMD patients. Together, these results support the existence of a variable effect of monocytes/macrophages function, especially concerning M2a macrophages, between individuals affected by AMD and unaffected individuals.

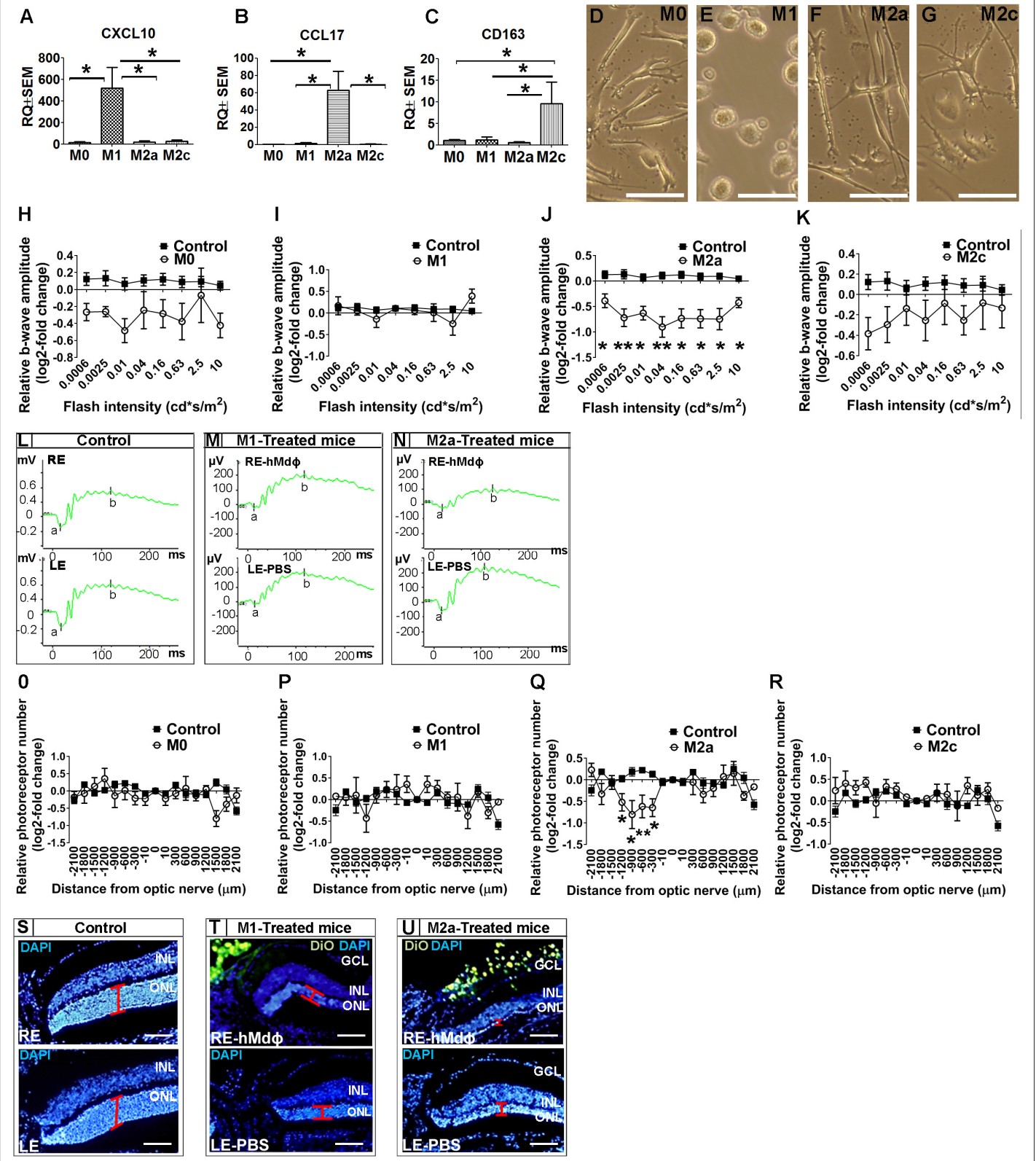

**Figure 1.** In vitro characterization of hMdφs and in vivo cytotoxicity of M2a hMdφ cells. (**A–C**) Human M0 macrophages (hMdφs) were polarized to form M1, M2a, or M2c hMdφs, and expression of the phenotype-specific markers CXCL10 (**A**), CCL17 (**B**), and CD163 (**C**), respectively, was confirmed using quantitative real-time PCR (n=5 per group, one-way ANOVA with multiple comparisons). (**D–G**) Inverted phase-contrast microscopy of M0 (**D**), M1 (**E**), M2a (**F**), and M2c (**G**) hMdφs, showing distinct morphologies for each subtype. (**H–K**) Relative electroretinography (ERG) b-wave amplitude versus

*Figure 1 continued on next page*

*Figure 1 continued*

light flash intensity of mice injected with M0 (**H**), M1 (**I**), M2a (**J**), and M2c (**K**) hMdɸs and control mice that were not exposed to light; ERG recordings revealed that adoptive transfer of M2a hMdɸs—but no other macrophage phenotypes—lead to suppressed b-wave amplitude (n=8 mice for each group, one-way ANOVA with multiple comparisons). The relative ERG b-wave was calculated by dividing the b-wave amplitude recorded from the mouse eye injected with hMdɸs by the b-wave amplitude recorder from the vehicle-injected eye of the same mouse. Similar b-wave amplitudes were recorded in fellow eyes of control mice. (**L–N**) Representative ERG recordings in control mice and in photic-injured mice in which the right eye (RE) was injected with M1(**M**) or M2a (**N**) hMdɸs and the left eye (LE) was injected with phosphate-buffered saline (PBS) as vehicle. 'a' and 'b' in the graph indicate the a-wave and the b-wave of the ERG, respectively. (**O–R**) Summary of the relative number of photoreceptor nuclei in the ONL measured at the indicated distances from the optic nerve head. A decrease in the number of photoreceptors nuclei was observed after adoptive transfer of M2a hMdɸ (**Q**), but not of other macrophage subtypes (**O, P, and R**) (n=8 mice for each group, one-way ANOVA with multiple comparisons). The relative number of photoreceptor nuclei was calculated by comparing the number of photoreceptor nuclei present in the ONL of the mouse eye injected with hMdɸ and the counterpart vehicle-injected eye. (**S–U**) Representative immunofluorescence images of retinal slices prepared from the indicated mice following an injection of DiO-stained hMdɸs (green); the nuclei were counterstained with DAPI (blue), and the ONL is indicated (red brackets). Note the presence of DiO-positive M2a hMdɸs in the GCL (**U**). GCL, ganglion cell layer; INL, inner nuclear layer; ONL, outer nuclear layer. Data shown as mean ± SEM. p-Values indicated by *p<0.05 and **p<0.01. Scale bars: 50 µm (**D–G and S–U**).

The online version of this article includes the following source data and figure supplement(s) for figure 1:

**Source data 1.** Real-time quantitative PCR (qPCR) analysis of *CXCL10*, *CCL17*, and *CD163* mRNA in M0, M1, M2a, and M2c macrophages derived from human monocytes (hMdɸs).

**Source data 2.** Electroretinography (ERG) b-wave recordings and outer nuclear layer (ONL) thickness of hMdɸs-treated mice and untreated mice.

**Figure supplement 1.** Establishing a model of photic retinal damage.

**Figure supplement 1—source data 1.** Electroretinography (ERG) b-wave recordings and outer nuclear layer (ONL) thickness of monocytes-treated mice and untreated mice.

**Figure supplement 2.** Adoptive transfer of M2a hMdɸ cells into mouse eyes.

In an attempt to explain the increased death of photoreceptors in the dorso-central retina following injection of M2a hMdɸs, after inducing photic injury we monitored the spatial distribution of the injected cells for 7 days using histology. We identified the injected M2a hMdɸs by their typical elongated cell shape (*Hagbi-Levi et al., 2017*; *McWhorter et al., 2013*) and the Dio tracer (*Figure 1—figure supplement 2B, D*). Although most of the injected M2a hMdɸs were scattered throughout the vitreous (*Figure 1—figure supplement 2C*), several of these cells migrated across the retina layers, reaching the subretinal space (*Figure 1—figure supplement 2G*), with many hMdɸ cells present around the ONH and along the retinal vessels (*Figure 1—figure supplement 2I, J*). A similar pattern of distribution was observed in eyes injected with the other types of hMdɸs (*Figure 1—figure supplement 2E, F and H*), indicating that the detrimental effect attributed to M2a phenotype was not related to its migration capacity.

Histological analysis revealed that the deleterious effects of M2a hMdɸs occurred primarily in the dorso-central region ranging at distances of –300 µm to –1200 µm from the ONH (*Figure 1Q*). A retinal flat-mount analysis and in vivo fundus autofluorescence confirmed that the injected M2a hMdɸs were found primarily in the superior half of the eye, corresponding to the area of photoreceptor cell death (*Figure 1—figure supplement 2K, L*).

## M2a hMdɸs have a neurotoxic effect on retinal tissue ex vivo

Our in vivo experiments showed that M1 hMdɸs did not induce neurotoxicity, whereas M2a hMdɸs had a robust neurotoxic effect. To support these in vivo findings, we compared the effect of M1 and M2a hMdɸs ex vivo by co-culturing retinal explants with $10^5$ cells for 18 hr, followed by TUNEL staining. We measured the number of apoptotic photoreceptor cells in the retinal explants using confocal microscopy (*Figure 2A–C*), and we performed cell sorting (*Figure 2D–G*). We found that explants co-cultured with M2a hMdɸs had significantly more apoptotic photoreceptor cells compared to both control retinal explants (without co-cultured macrophages) and explants co-cultured with M1 hMdɸs (*Figure 2H*). We then determined the cell types that were affected by the M2a macrophages in the retinal explants and in choroid-RPE explants that were co-cultured with M2a hMdɸs, revealing photoreceptor cell death (*Figure 2I*) and RPE cell death (*Figure 2J and K*), respectively.

Although the injected M2a hMdɸs were observed in different retinal layers in our in vivo experiments, it is also possible that mediators released from these hMdɸs—and not necessarily direct contact with the macrophages themselves—contributed to the increase in photoreceptor cell death.

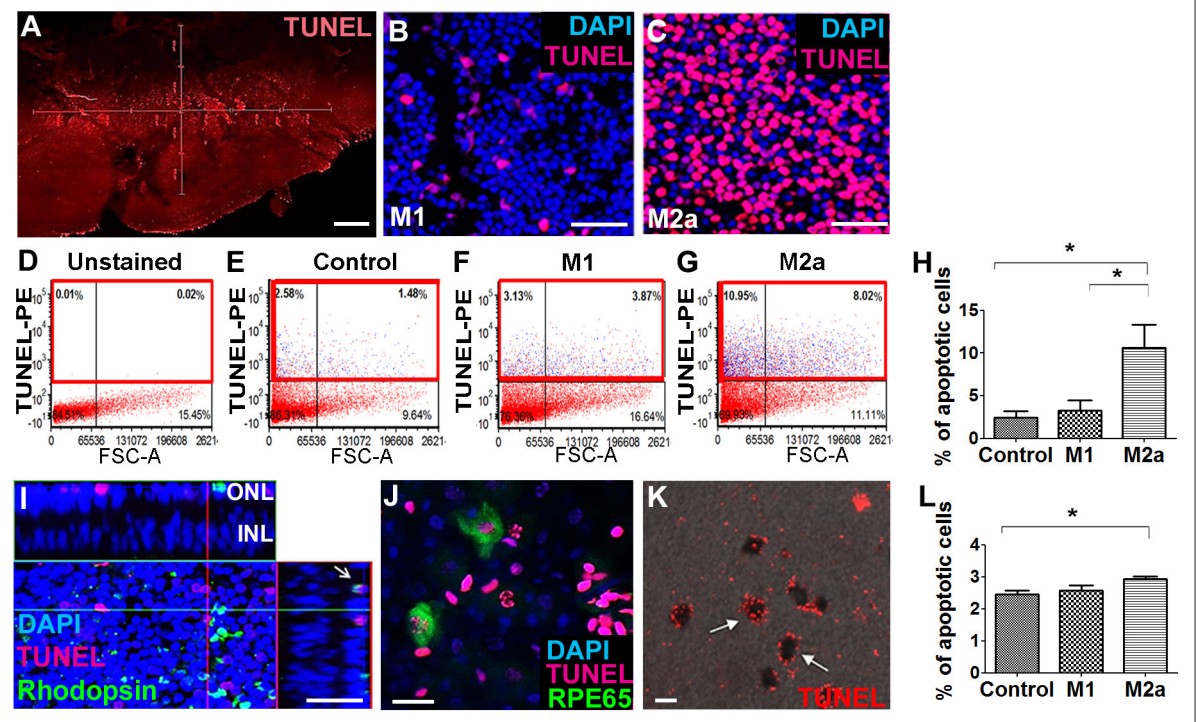

**Figure 2.** M2a hMdφs have neurotoxic effects on retinal explants. (**A–C**) TUNEL staining (red) of mouse retinal explants co-cultured with either M1 hMdφs (**B**) or M2a hMdφs (**C**) for 18 hr; the nuclei were counterstained with DAPI (blue). Confocal microscopy assessment of 11 randomly selected retinal fields (**A**) revealed M2a hMdφ (**C**) to be associated with photoreceptor cell apoptosis as compared to M1 hMdφ. (**D–G**) Representative FACS gating plot of cells obtained from the indicated retinal explants stained using TUNEL-PE; TUNEL⁺ cells are shown in the red rectangles based on an unstained retinal explant. (**H**) Summary of the percentage of apoptotic (TUNEL⁺) cells measured in control explants and in explants that were co-cultured with M1 or M2a hMdφs (n=6 per group, one-way ANOVA with multiple comparisons). (**I**) Rhodopsin immunostaining (green) of a retinal explant showing the association of macrophages with apoptotic photoreceptor cells, indicated by the co-localization of TUNEL and rhodopsin staining in a Z-stack (arrow). (**J–K**) Co-culturing retinal pigment epithelial (RPE)-choroid explants with M2a hMdφs results in the co-localization of TUNEL staining (red) and the RPE marker RPE65 marker (J; green), as well as the presence of TUNEL-positive hexagonal and pigmented cells (K; arrows). (**L**) The addition of supernatant collected from M2a hMdφs to retinal explants increased photoreceptor death (measured as an increase in the percentage of TUNEL-positive cells) (n=6 per group, one-way ANOVA with multiple comparisons). INL, inner nuclear layer; ONL, outer nuclear layer. Data shown as mean ± SEM. p-Values indicated by *p<0.05. Scale bars: 20 μm (**G and H**), 50 μm (**B, C, and F**), and 200 μm (**A**).

The online version of this article includes the following source data for figure 2:

**Source data 1.** Summary of the percentage of apoptotic (TUNEL⁺) cells measured in the different groups of retinal explant.

To determine whether direct contact between M2a hMdφs and photoreceptor cells is required for inducing apoptosis, we examined retinal explants that were treated with cell-free hMdφ-conditioned medium. We found that retinal explants cultured with supernatant from M2a hMdφs had significantly increased cell death; in contrast, culturing explants with supernatant from M1 hMdφs had no effect (*Figure 2L*). Interestingly, retinal explants co-incubated with M2a hMdφ cells had a higher percentage of apoptosis (10%; *Figure 2H*) compared to explants incubated with M2a hMdφ-conditioned medium (3%; p=0.028; *Figure 2L*). These ex vivo results suggest that M2a hMdφ cells have the capacity to directly affect neuronal tissues independent of the systemic inflammatory context and without the need to recruit additional cell types.

## Characterization of the neurotoxic properties of M2a hMdφs

Next, we attempted to characterize the putative neurotoxic effects of M2a hMdφs on photoreceptor cells. Macrophages are a potential source of reactive oxygen species (ROS), and oxidative stress has been implicated in the progression of various retinal diseases, including AMD (*Beatty et al., 2000*). To explore the role of ROS production in M2a hMdφ-mediated neurotoxicity, we measured in vitro ROS production in M0, M1, and M2a hMdφs and found that M2a hMdφs release significantly more ROS compared to both M0 and M1 hMdφs (*Figure 3A*). We also used hydroxynonenal (HNE) staining

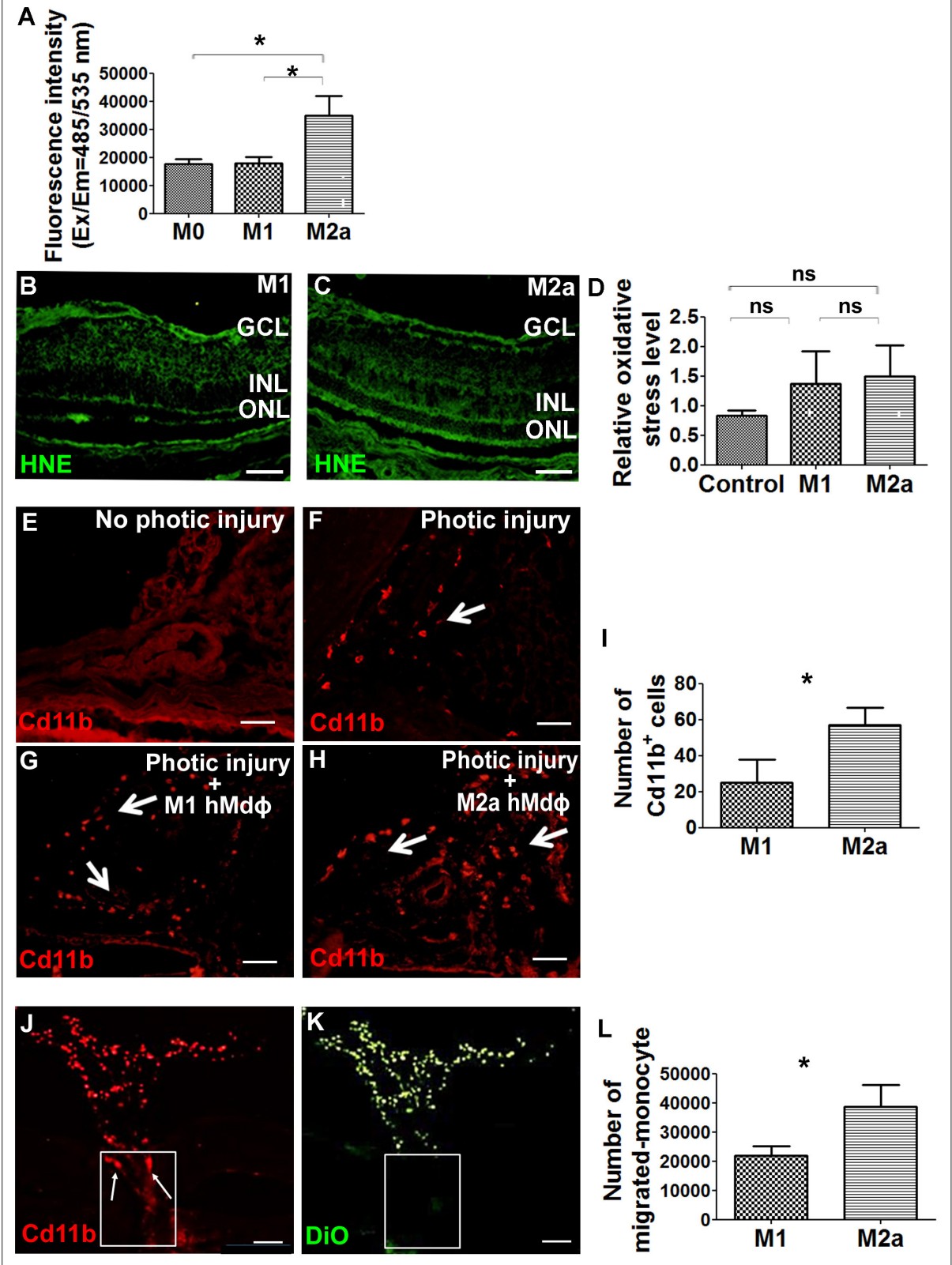

**Figure 3.** M2a hMdɸs produce high levels of reactive oxygen species (ROS) and induce the infiltration of cd11b⁺ cells. (**A**) Cultured M2a hMdɸs produce higher levels of ROS than both M0 and M1 hMdɸs; ROS production was measured using the DCFDA fluorogenic dye (n=5 per group, one-way ANOVA with multiple comparisons). (**B–C**) Hydroxynonenal (HNE) immunostaining (green) shows similar levels of oxidative stress in mouse eyes following an intravitreal injection of either M1 hMdɸs (**B**) or M2a hMdɸs (**C**). (**D**) Summary of relative oxidative stress levels measured in control eyes and in eyes

*Figure 3 continued on next page*

*Figure 3 continued*

injected with M1 or M2a hMdɸs calculated by comparing the fluorescence level of HNE-stained sections of the hMdɸ-injected eye and the vehicle-injected eye of the same mouse. HNE staining intensity was quantified using ImageJ (n=7 per group, one-way ANOVA with multiple comparisons). (**E–H**) cd11b immunostaining (red, arrows) shows an increased presence of mononuclear phagocytes in the choroid tissue following photic injury (**F**) and in eyes following an intravitreal injection of M1 hMdɸs (**G**) or M2a hMdɸs (**H**). (**I**) Summary of cd11b+ cells present in the choroid tissue in eyes following an intravitreal injection of M1 or M2a hMdɸs (n=7 per group, Student's t-test). (**J–K**) Inverted phase-contrast microscopy images of the optic nerve head (ONH) showing the migration of DiO-labeled M2a hMdɸ cells (K; green) and co-localization with recruited cd11b+ cells (J; red), shown in the rectangles. (**L**) An in vitro migration assay was performed using a Boyden chamber, showing an increase in monocytes that migrated toward the M2a hMdɸs compared with the M1 hMdɸs (n=6 per group, Student's t-test). GCL, ganglion cell layer; INL, inner nuclear layer; ONL, outer nuclear layer. Data shown as mean ± SEM. p-Values indicated by ns, nonsignificant, *p<0.05. Scale bars: 50 μm (**B–C and E–H**) and 100 μm (**J–K**).

The online version of this article includes the following source data for figure 3:

**Source data 1.** Comparative analysis of reactive oxygen species (ROS) production, fluorescence level of hydroxynonenal (HNE)-stained sections, cd11b-positive cells quantification, and in vitro migration level between M1 and M2a hMdɸs.

to evaluate the oxidative damage in a retinal section following photic injury and found that injection of either M1 or M2a hMdɸs was not associated with increased oxidative damage compared to control conditions (*Figure 3B, C and D*). Exposing retina to strong light (such as in photic retinal injury) leads to the ROS production due to exacerbated action of the visual cycle (*Ozawa, 2020*). Thus, the additional limited contribution of ROS production by M2a hMdɸs in photic injured retina may be difficult to detect. Together, these data suggest that M2a hMdɸ-mediated neurotoxicity may be partially also driven by increased ROS release from these cells, although further investigations are required to confirm this possibility.

Although the ex vivo results obtained from our retinal explant experiments suggest that M2a hMdɸs exert a direct neurotoxic effect, additional indirect processes may also contribute to this effect in vivo. For example, the presence of M2a hMdɸs may drive the recruitment of mononuclear cells to the retina, and these cells may exert an additional neurotoxic effect. To examine this possibility, we measured cells expressing cd11b—a broadly expressed integrin that serves as a marker of mononuclear phagocytes—in the choroid of photic-injured mice following an injection of either M1 or M2a hMdɸs. We found that injecting M2a hMdɸs led to significantly more recruitment of cd11b+ cells to the choroid compared to injecting M1 hMdɸs (*Figure 3G–I*). In addition, we found that M2a hMdɸs and cd11b+ cells were co-localized across the ONH, suggesting the existence of crosstalk between these two cell types (*Figure 3J and K*).

In principle, an inflammatory response may have potentially resulted from the xenograft; however, the eye is an immune-privileged site, and we previously excluded the possibility that adoptive transfer of hMdɸs causes a cross-species reaction (*Hagbi-Levi et al., 2017*). In addition, we tested all four hMdɸ subtypes in our xenograft model but found that only M2a hMdɸs were associated with an increased recruitment of endogenous cells. To confirm that cd11b+ cells are indeed recruited specifically by M2a hMdɸs, we measured the in vitro chemotactic capacity of M1 and M2a hMdɸs on freshly isolated human monocytes. Using FACS analysis, we found that chemokines released from M2a hMdɸs attracted more monocytes compared to chemokines released from M1 hMdɸs (*Figure 3L*). Taken together, these results indicate that M2a hMdɸs differ from the M1 hMdɸs phenotypes with respect to their capacity to recruit additional immune cells to the site of injury and their ability to increase oxidative stress, thereby exacerbating photoreceptor cell death in the context of inflammation.

## CCR1 expression and apoptosis are increased in the retina following photic-induced damage

Next, we examined whether a cytokine-mediated interaction between M2a hMdɸs and the retinal environment underlies photoreceptor cell death. Using a multiplex cytokine array, we compared the levels of 120 cytokines (*Supplementary file 1*) between M1 hMdɸ-conditioned medium and M2a hMdɸ-conditioned medium (n=6 per group, Student's t-test).

We found that 9 cytokines were significantly higher in the M2a hMdɸ-conditioned medium, while 15 cytokines were significantly higher in the M1 hMdɸ-conditioned medium (Table 2). Several of the 9 cytokines that were increased in the M2a hMdɸ-conditioned medium were previously reported to play a role in various inflammatory processes, including ocular inflammatory diseases and neurodegenerative diseases; these cytokines include CCL11 (*Deng et al., 2019*; *Mor et al., 2019*; *Segal-Salto et al.,*

*2020*; *Shoji et al., 2017*; *Zeng et al., 2019*), CCL13 (*El-asrar et al., 2019*; *Méndez-Enríquez et al., 2014*; *Mendez-Enriquez and García-Zepeda, 2013*; *Zeng et al., 2019*), CCL17 (*Dai et al., 2015*; *Yu et al., 2011*; *Zeng et al., 2019*), CCL23 (*Faura et al., 2020*; *Simats et al., 2018*; *Zeng et al., 2019*), and CCL14 (*Liu et al., 2016*).

Interestingly, three of these cytokines (CCL14, CCL13, and CCL23) are ligands of the C-C chemokine receptor CCR1. To confirm these multiplex ELISA results, we performed real-time quantitative PCR (qPCR) analysis of the mRNAs that encode these three CCR1 ligands and found significantly higher levels of both *CCL23* and *CCL13* mRNA in M2a hMdΦs compared to M1 hMdΦs (*Figure 4—figure supplement 1A, B*); in contrast, we found no significant difference in *CCL14* mRNA levels between M2a and M1 hMdΦs (*Figure 4—figure supplement 1C*).

Next, we attempted to identify which cell type(s) in the retina express CCR1 and are therefore affected by the cytokines released by M2a hMdΦs and drive photoreceptor cell death in response to photic-induced injury. Using immunofluorescence, we found increased levels of CCR1 protein in the mouse retina—primarily in the ONL— 48 hr after inducing photic injury (*Figure 4B*). We also measured robust CCR1 immunofluorescence in the inner nuclear layer (INL) and inner plexiform layer (IPL) 7 days after photic injury (*Figure 4C*). We then performed dual immunostaining for CCR1 and the glial cell marker GFAP in retinal sections and found strong co-localization of these two proteins following photic injury (*Figure 4K*). These results indicate that CCR1 is expressed primarily in Müller cells, the only retinal cell type that spans all of the layers of the retina (*Bringmann et al., 2006*).

Using TUNEL staining, we found a large number of apoptotic photoreceptor cells 48 hr after photic injury (*Figure 4E*); 7 days after photic injury, apoptotic photoreceptor cells were still present (*Figure 4F*), albeit it to a lesser extent as previously reported (*Jiao et al., 2015*). To confirm that photic injury increases the expression of CCR1 in the retina, we measured *Ccr1* mRNA using qPCR 7 days after photic injury and found significantly increased retinal expression of *Ccr1* compared to control mice (*Figure 4G*). We also measured retinal function using ERG recordings and found a strong inverse correlation between retinal *Ccr1* mRNA levels and b-wave amplitude following photic injury (*Figure 4H*), suggesting that *Ccr1* expression may play a role in determining the extent of retinal damage.

Previous studies suggest that CCR1, CCR2, and CCR5 may be functionally redundant (*Gladue et al., 2010*). We therefore examined whether photic injury also increases the level of CCR2 and/or CCR5 protein in Müller cells in the mouse retina using immunofluorescence. Interestingly, however, neither CCR2 (*Figure 4—figure supplement 2B*) nor CCR5 (*Figure 4—figure supplement 2D*) was detected in these cells in either control or photic-injured mice.

Consistent with previous reports that photic injury can promote the recruitment of inflammatory cells (*Lückoff et al., 2017*)—particularly macrophages that express CCR1, CCR2, and CCR5 (*Rutar et al., 2015*)—we found that cells expressing CCR1, CCR2, and CCR5 were recruited to the subretinal space in photic-damaged mice (*Figure 4—figure supplement 2E–G*); moreover, we measured increased levels of both *Ccr2* (*Figure 4—figure supplement 2H*) and *Ccr5* (*Figure 4—figure supplement 2I*) mRNA following photic injury. Taken together, these results suggest that the increased expression of CCR2 and CCR5 following photic injury stems primarily from immune cells that were recruited to the site of damage, while the increased expression of CCR1 likely stems from both increased expression in Müller cells and the recruitment of inflammatory cells to the damaged retina.

## Increased expression of CCR1 in rd10 mice and senescent mice

Next, we examined whether CCR1 is also upregulated in other models of retinal degeneration. The rd10 mouse is a model of autosomal recessive retinitis pigmentosa in which a mutation in the *Pde6b* gene (which encodes the enzyme phosphodiesterase in rod cells) causes degeneration of photoreceptor cells starting at around postnatal day 18 (*Chang et al., 2002*). At 7 days of age—that is, before the onset of photoreceptor cell apoptosis—we measured extremely low levels of CCR1 in the retina (*Figure 5A*). In contrast, we measured significantly higher expression of CCR1 in both the INL and ONL at 3 and 6 weeks of age, together with a reduction in ONL thickness of approximately 50% and 90% at 3 and 6 weeks, respectively (*Figure 5B and C*). Co-staining for CCR1 and GFAP at 3 weeks of age confirmed that the increased expression of CCR1 occurred specifically in Müller cells (*Figure 5I*).

qPCR analysis of *Ccr1* mRNA confirmed the increased expression of CCR1 in rd10 mice at both 3 and 6 weeks of age compared to 1 week (*Figure 5J*). Similarly, we also measured increased levels of

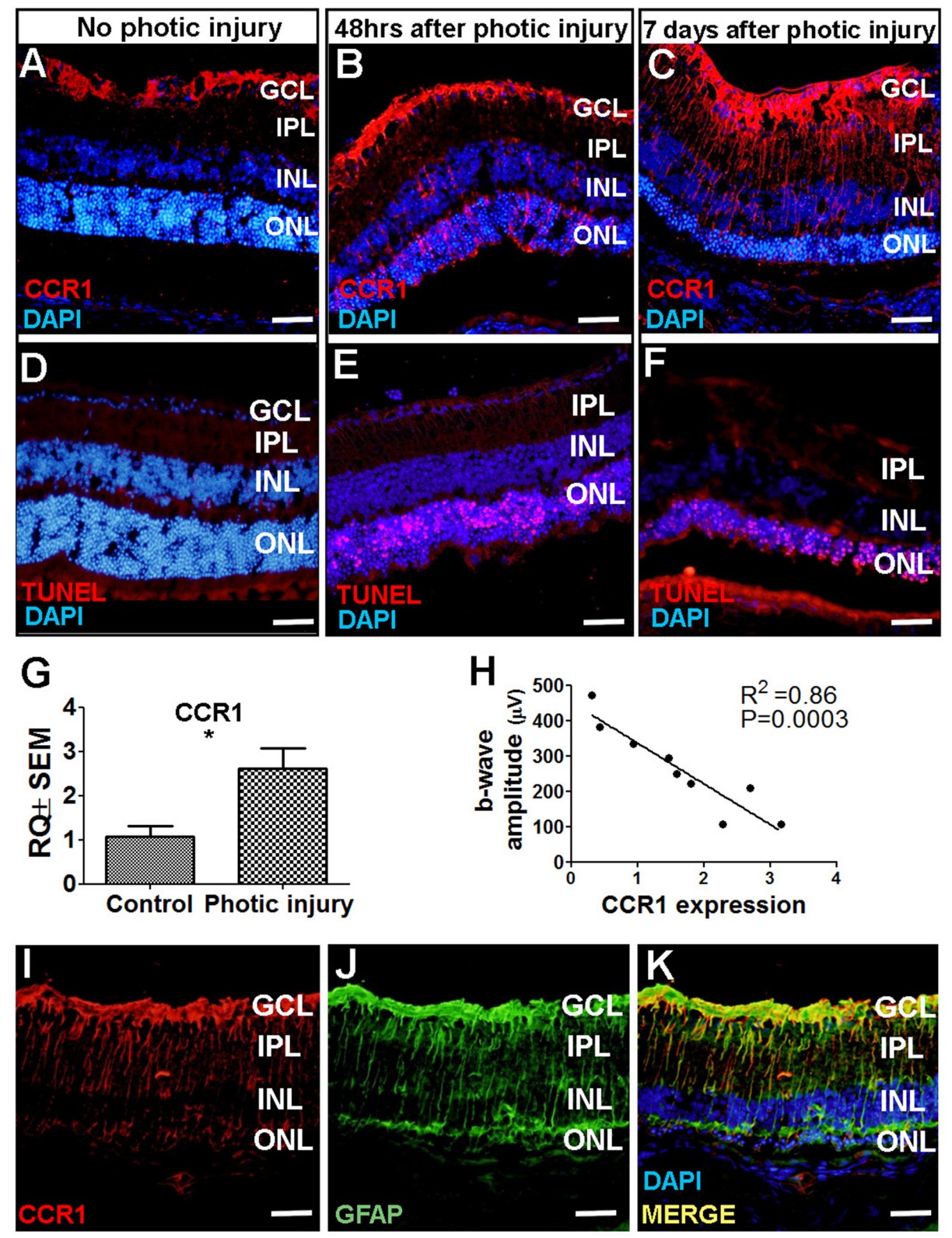

**Figure 4.** CCR1 is upregulated in a mouse model of photic injury. (**A–F**) Albino BALB/c mice were exposed to photic injury; 48 hr and 7 days later, retinal sections were prepared and immunostained for Ccr1 (A–C; red) or TUNEL stained (D–F, red); the nuclei were counterstained with DAPI (blue). (**G**) Real-time quantitative PCR (qPCR) analysis of retinal *Ccr1* mRNA measured in control mice and in mice 7 days after photic injury (n=6 mice for each group, Student's t-test). (**H**) *Ccr1* expression plotted against the b-wave amplitude measured using electroretinography (ERG) in mice 7 days after photic injury.

*Figure 4 continued on next page*

*Figure 4 continued*

Each symbol represents an individual mouse eye, and the correlation coefficient and p-value are shown. (**I–K**) Retinal sections were prepared 7 days after photic injury and co-immunostained for CCR1 (I; red) and the glial cell marker GFAP (J; green); the nuclei were counterstained with DAPI (blue). GCL, ganglion cell layer; INL, inner nuclear layer; IPL, inner plexiform layer; ONL, outer nuclear layer. Data shown as mean ± SEM. p-Values indicated by **$p<0.005$. Scale bars: 50 µm (A–F and I–K).

The online version of this article includes the following source data and figure supplement(s) for figure 4:

**Source data 1.** Real-time quantitative PCR (qPCR) analysis of retinal *Ccr1* mRNA in control mice vs. photic-injured mice and the correlation between the *Ccr1* expression and the electroretinography (ERG) b-wave amplitude.

**Figure supplement 1.** Two CCR1 ligands are expressed at higher levels in M2a hMdɸs compared to M1 hMdɸs.

**Figure supplement 1—source data 1.** Real-time quantitative PCR (qPCR) analysis of *CCL23*, *CCL13*, and *CCL14* mRNA in M1 and M2a macrophages derived from human monocytes.

**Figure supplement 2.** Expression of CCR2 and CCR5 in the mouse retina following photic injury.

**Figure supplement 2—source data 1.** Real-time quantitative PCR (qPCR) analysis of retinal *Ccr2* and *Ccr5* mRNA measured in control mice and photic-injured mice.

both *Ccr2* and *Ccr5* mRNA at 3 weeks of age compared to 1 week (*Figure 5K*); however, neither CCR2 nor CCR5 was present in Müller cells based on immunohistochemistry (*Figure 4—figure supplement 2J–L*). Together, these results support the notion that CCR1 expressed in Müller cells plays a distinct role in photoreceptor cell death.

As the name suggests, AMD primarily affects the elderly (*Wong et al., 2014*). Interestingly, we found that 18-month-old wild-type BALB/c mice (i.e., 'elderly' or senescent mice) express CCR1 in both the INL and ONL (*Figure 5L*), and co-immunostaining for CCR1 and GFAP shows that CCR1 is expressed primarily in Müller cells (*Figure 5M and N*). Moreover, qPCR analysis confirmed that senescent mice have increased levels of retinal *Ccr1* mRNA compared to young mice (*Figure 5O*); in contrast, we found no difference between senescent and young mice with respect to *Ccr2* or *Ccr5* mRNA levels (*Figure 5O*), and immunohistochemistry confirmed that neither CCR2 nor CCR5 is expressed in Müller cells (unpublished observations).

## Inhibiting CCR1 reduces photic injury-induced retinal damage

Based on our finding that CCR1 expression is upregulated in the retina in: (i) photic-injured mice, (ii) rd10 mice in parallel with the onset of retinal degeneration, and (iii) senescent mice, we hypothesized that inhibiting this receptor may slow the rate of photoreceptor loss. To test this hypothesis, we injected mice with the CCR1-specific inhibitor BX471 (or vehicle in control mice) immediately after inducing photic injury. BX471 is a non-peptide antagonist that can bind CCR1 and blocks its signal transduction (*Liang et al., 2000*). We found that BX471-treated mice had both a larger b-wave amplitude on ERG (*Figure 6A*) and increased ONL thickness compared to vehicle-treated mice (*Figure 6B*), suggesting that inhibiting CCR1 can help against photic injury in mice.

Interestingly, immunostaining for the protein IBA-1 (ionized calcium-binding adaptor molecule 1, a marker of microglial activation) revealed that photic injury-induced microglial activation was reduced in BX471-treated mice. Specifically, we found that BX471-treated mice photic-injured had elongated microglial cells that were localized primarily to the ganglion cell layer and INL (*Figure 6D*); in contrast, vehicle-treated photic-injured mice had amoeboid-shaped microglial cells that infiltrated both the ONL and subretinal space (*Figure 6C*). Moreover, qPCR analysis revealed that photic injury increased the recruitment of macrophages to the retina (based on increased retinal expression of the macrophage marker F4/80), and this recruitment was significantly reduced in BX471-treated mice (*Figure 6E*). We also found that BX471 reduced CCR1 expression in Müller cells in photic-injured mice compared to control-treated mice (*Figure 6G*); this finding was confirmed using qPCR to measure *Ccr1* mRNA (*Figure 6H*).

Our finding that CCR1 is expressed primarily in Müller cells suggests that this receptor may play a key functional role in these cells. We therefore measured whether three genes encoding markers of activated Müller cells—namely, *Ccl2*, *Cxcl1*, and/or *Cxcl10*—are upregulated following photic injury. Consistent with the previous reports (*Natoli et al., 2017*; *Rutar et al., 2015*), we found increased expression of all three genes following photic injury (*Figure 6I–K*). In addition, treating mice with the CCR1 inhibitor BX471 reduced expression to control levels (*Figure 6I–K*). Taken together, these

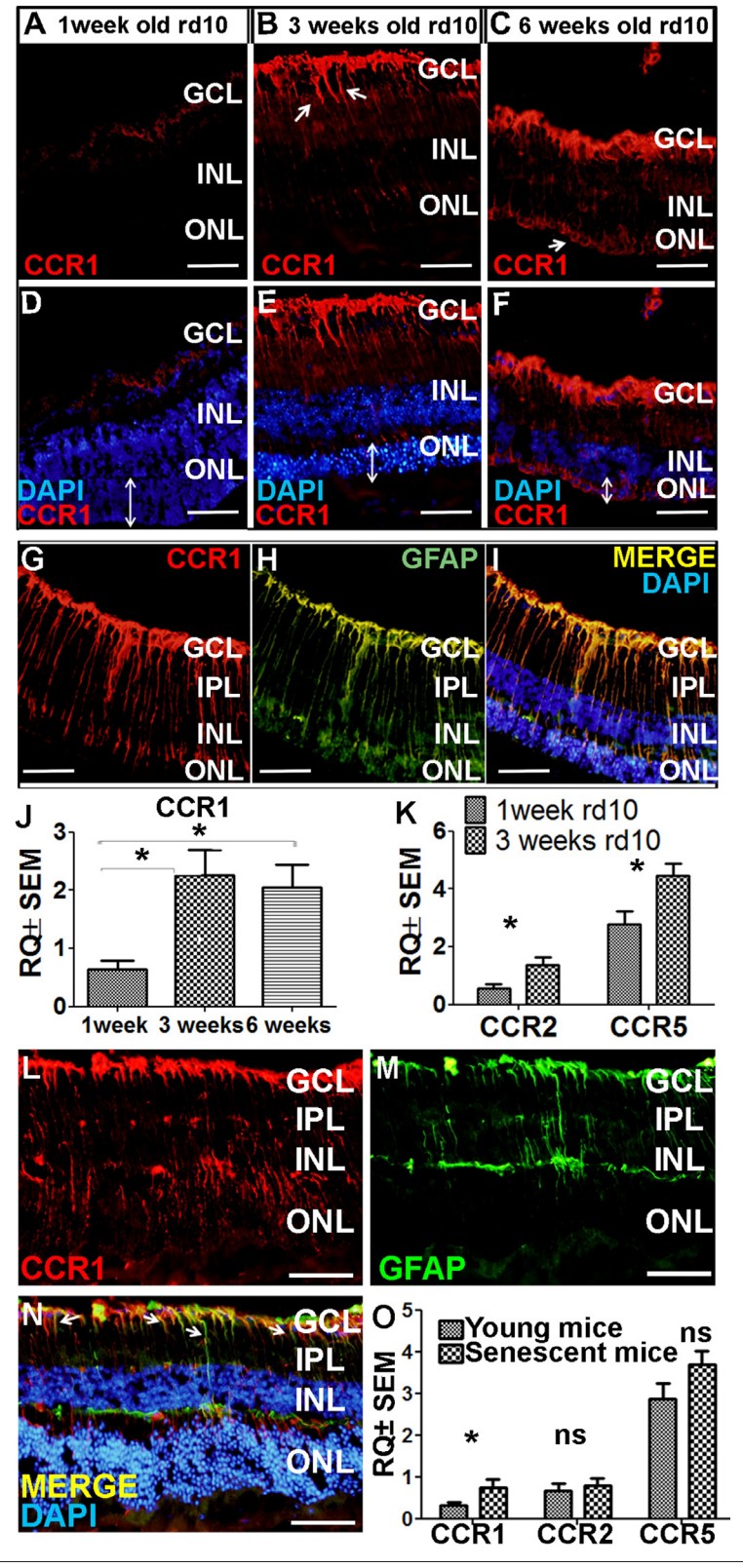

**Figure 5.** Ccr1 is upregulated in both rd10 mice and senescent mice. (**A–F**) Retinal sections were prepared from 1-week-old (**A, D**), 3-week-old (**B, E**), and 6-week-old (**C, F**) rd10 mice and immunostained for CCR1; the nuclei were counterstained with DAPI. Note the increased expression of CCR1 (arrows) at 3 weeks (**B**) and 6 weeks (**C**), corresponding with reduced ONL thickness (E and F; double-ended arrows). (**G–I**) Retinal sections were

*Figure 5 continued on next page*

*Figure 5 continued*

prepared from a 3-week-old rd10 mouse and co-immunostained for CCR1 (G; red) and GFAP (H; green); the nuclei were counterstained with DAPI (**I**). Note the co-localization of CCR1 and GFAP in the Müller cells (**I**). (**J**) Real-time quantitative PCR (qPCR) analysis of retinal *Ccr1* mRNA measured in 1-, 3-, and 6-week-old rd10 mice (n=6 mice for each group, one-way ANOVA with multiple comparisons). (**K**) Real-time qPCR analysis of retinal *Ccr2* and *Ccr5* mRNA measured in 1- and 3-week-old rd10 mice (n=4 mice for each group, Student's t-test). (**L–N**) Retinal sections were prepared from 18-month-old (senescent) mice and co-immunostained for CCR1 (L; red) and GFAP (M; green); the nuclei were counterstained with DAPI. Note the co-localization of CCR1 and GFAP in the Müller cells (N; arrows). (**O**) Real-time qPCR analysis of retinal *Ccr1*, *Ccr2*, and *Ccr5* mRNA measured in young (6-week-old) and senescent (18-month-old) mice (n=8 mice for each group, Student's t-test). GCL, ganglion cell layer; INL, inner nuclear layer; IPL, inner plexiform layer; ONL, outer nuclear layer. Data shown as mean ± SEM. p-Values indicated by ns, not significant and *p<0.05. Scale bars: 50 μm (**A–I and L–N**).

The online version of this article includes the following source data for figure 5:

**Source data 1.** Real-time quantitative PCR (qPCR) analysis of retinal *Ccr1*, *Ccr2,* and *Ccr5* mRNA measured in 1-, 3-, 6-week-old rd10 mice, in young (6-week-old) and senescent (18-month-old) mice.

results indicate that inhibiting CCR1 can reduce the retinal inflammation induced by photic injury and increase the survival of photoreceptor cells.

## Inhibiting CCR1 reduces the neurotoxic effects of M2a macrophages

Given that CCR1 is a chemokine receptor expressed by a wide range of immune cells, including mononuclear cells (*Gao et al., 1993*; *Mantovani et al., 2006*), we asked whether inhibiting this receptor can affect the functional properties of M2a hMdɸs via an autocrine signaling process. We found that both M1 and M2a hMdɸs express CCR1 (*Figure 7A and B*); however, cell sorting analysis revealed that a significantly larger percentage of M2a hMdɸs express CCR1 compared to M1 hMdɸs (*Figure 7D*), suggesting that M2a hMdɸs may be more susceptible to the effects of inhibiting CCR1. We therefore examined whether inhibiting CCR1 could reduce M2a hMdɸ-mediated neurotoxicity and found that treating M2a hMdɸs with either 0.5 μM or 5 μM BX471 significantly reduced their production of ROS (*Figure 7E*). In addition, treating monocytes from patients with AMD with 10 μM BX471 significantly reduced the ability of M2a to attract monocytes (*Figure 7F*), indicating that the recruitment of mononuclear cells by M2a hMdɸs is mediated in part by CCR1 signaling.

## Discussion

The role of monocytes—and macrophages in particular—in the pathogenesis of AMD has received increasing attention in recent years. Here, we provide additional insights into the function of specific hMdɸ phenotypes in the context of aAMD. We found that M2a hMdɸs mediate neurotoxicity in both in vitro and in vivo models of aAMD. In addition, and contrary to the prevailing hypothesis that M1 macrophages likely underlie tissue damage during inflammation, we found that M1 macrophages do not appear to play a major role in retinal damage in the context of aAMD. With respect to the potential underlying mechanism, we found that M2a hMdɸs produce high levels of ROS ex vivo; however, the in vivo effects of M2a hMdɸs may also be mediated by additional mechanisms such as increased production of cytokines that promote neurotoxicity and drive the recruitment of additional mononuclear cells. These findings may therefore explain the relatively high contribution of M2a macrophages to the pathogenesis of AMD. Indeed, oxidative stress—particularly ROS-induced cellular damage—was recently reported as a cause of retinal inflammation (*Abokyi et al., 2020*), and the recruitment of other immune cell types can exacerbate inflammation in the eye, an immune-privileged organ in which overstimulation of the immune system can be detrimental (*Buschini et al., 2011*).

By examining the molecular mechanism by which M2a hMdɸs drive retinal damage, we found that these cells can interact with retinal cells via the chemokine receptor CCR1 to mediate photoreceptor cell death. Although this receptor is expressed by a wide range of immune cell types and plays a key role in recruiting monocytes (*Trebst et al., 2002*), our results provide the first evidence that CCR1 is also expressed in Müller cells, and this expression increases during acute retinal damage (e.g., following photic injury) and during progressive retinal degeneration (e.g., in the rd10 mouse). We also found increased retinal expression of CCR1 in senescent mice, supporting the notion that this receptor

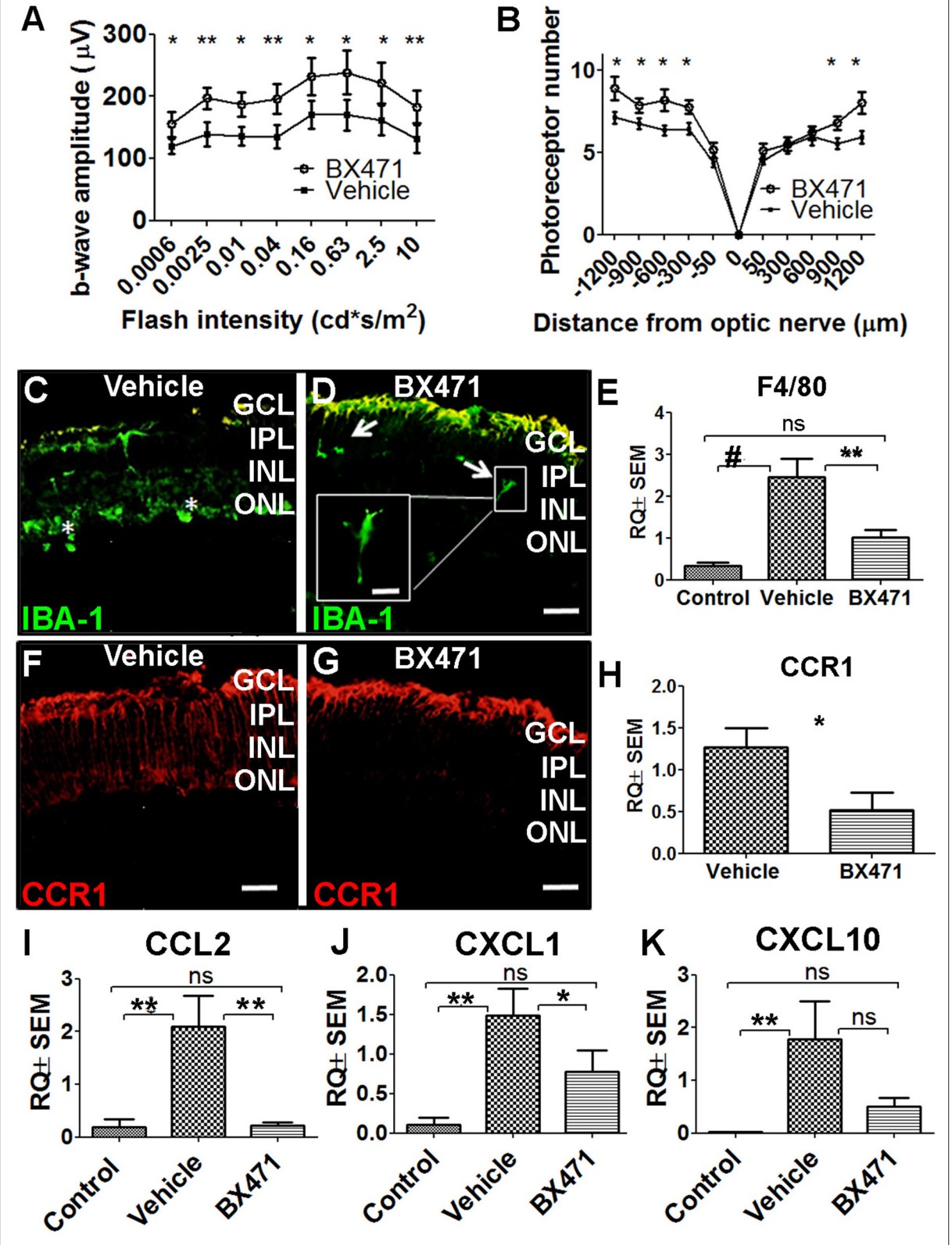

**Figure 6.** Inhibiting CCR1 reduces the effects of photic injury. (**A**) Albino BALB/c mice were subjected to photic injury followed by subcutaneous injections of the CCR1 inhibitor BX471 or vehicle for 5 days. Electroretinography (ERG) recordings were then performed, and the amplitude of the b-wave was measured and is plotted against flash intensity (n=12 eyes for each group, Student's t-test). (**B**) The number of photoreceptor nuclei was measured at the indicated distances from the optic nerve (n=14 eyes for each group, Student's t-test). (**C–G**) Retinal sections were prepared from

*Figure 6 continued on next page*

*Figure 6 continued*

vehicle-treated mice (**C**) and BX471-treated mice (**D**) and immunostained for the microglial cell marker IBA-1. The asterisks indicate amoeboid-shaped cells in the ONL and subretinal layer (**C**), and the arrows indicate elongated cells in the GCL and IPL, with one cell shown in a magnified view (D; inset). (**E**) Real-time quantitative PCR (qPCR) analysis of retinal *Adgre1* mRNA (which encodes the macrophage marker F4/80) in control mice, vehicle-treated photic-injured mice, and BX471-treated photic-injured mice (n=6 mice for each group, one-way ANOVA with multiple comparisons). (**F–G**) Retinal sections were prepared from vehicle-treated photic-injured mice (**F**) and BX471-treated photic-injured mice (**G**) and immunostained for CCR1. (**H**) Real-time qPCR analysis of retinal *Ccr1* mRNA measured in vehicle-treated photic-injured mice and BX471-treated photic-injured mice (n=6 mice for each group, Student's t-test). (**I–K**) Real-time qPCR analysis of retinal *Ccl2* (**I**), *Cxcl1* (**J**), and *Cxcl10* (**K**) mRNA measured in control mice, vehicle-treated photic-injured mice, and BX471-treated photic mice (n=6 mice for each group, one-way ANOVA with multiple comparisons). GCL, ganglion cell layer; INL, inner nuclear layer; IPL, inner plexiform layer; ONL, outer nuclear layer. Data shown as mean ± SEM. p-Values indicated by ns, not significant and *p<0.05, **p<0.01, #p<0.001. Scale bars: 50 μm (**C–D and F–G**) and 20 μm (inset in D).

The online version of this article includes the following source data for figure 6:

**Source data 1.** Electroretinography (ERG) b-wave recordings and outer nuclear layer (ONL) thickness of vehicle-treated mice and BX471-treated mice.

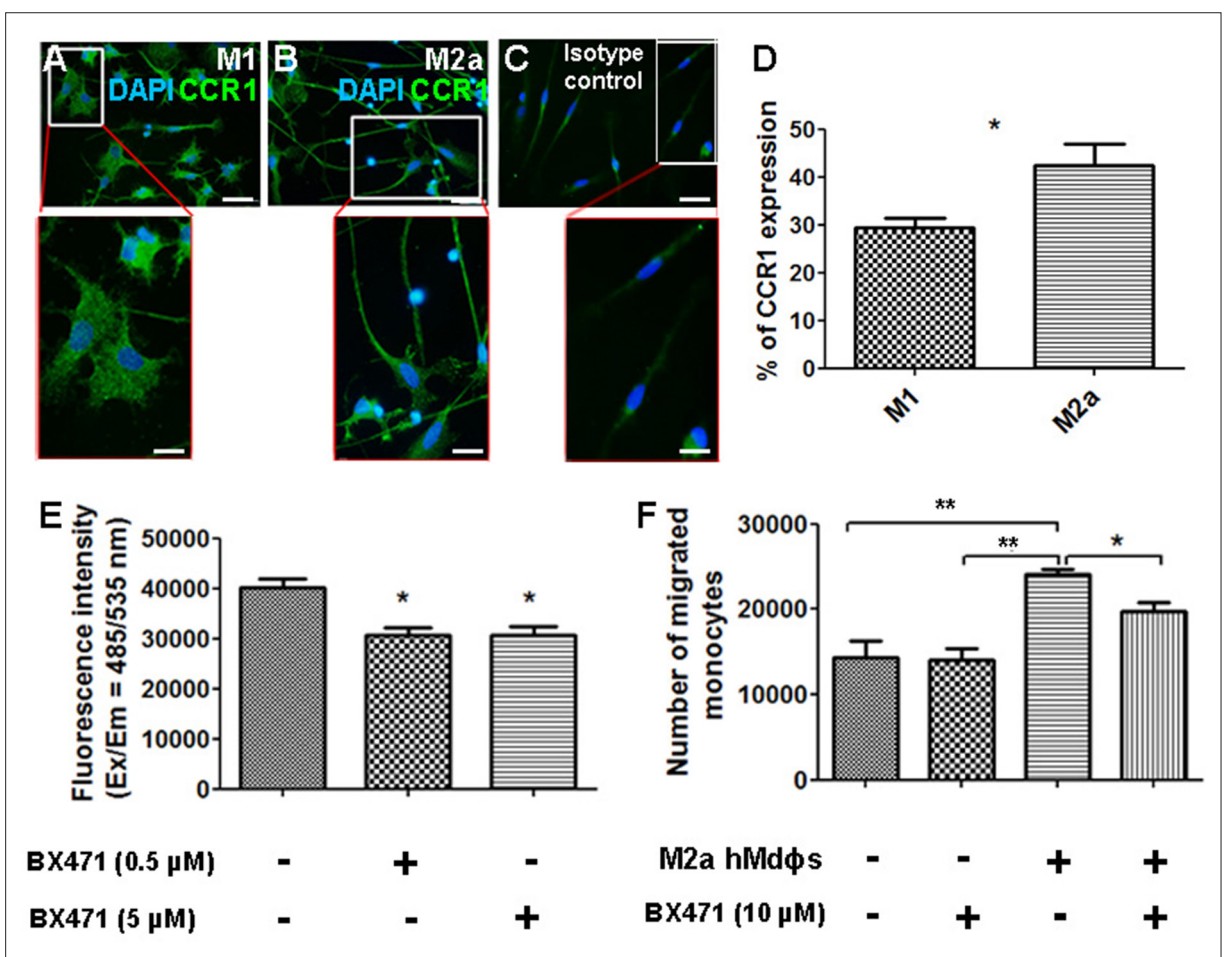

**Figure 7.** Inhibiting CCR1 modulates the functional properties of M2a hMdφs. (**A–C**) M1 (**A**) and M2a (**B**) hMdφs were immunostained for CCR1 (green); magnified views are shown below. (**D**) Summary of the percentage of CCR1-postive cells measured using cell sorting analysis of CCR1-stained M1 hMdφ and M2a hMdφs (n=5 per group, Student's t-test). (**E**) Summary of reactive oxygen species (ROS) levels measured in untreated M2a hMdφs and M2a hMdφs treated with 0.5 or 5 μM BX471 (n=4 per group, one-way ANOVA with multiple comparisons). (**F**) Summary of the migration of monocytes treated or not with 10 μM BX471 that migrated toward the M2a hMdφs using a Boyden chamber (n=3 per group, one-way ANOVA with multiple comparisons). Data shown as mean ± SEM. p-Values indicated by *p<0.05 and **p<0.01. Scale bars: 50 μm (**A–C**) and 20 μm (insets in A–C).

The online version of this article includes the following source data for figure 7:

**Source data 1.** Cell sorting analysis of CCR1-stained M1 hMdφ and M2a hMdφs.

is involved in age-related neurodegenerative diseases, including AMD. Finally, we found that inhibiting CCR1 significantly reduced the severity of retinal damage induced by photic injury, suggesting that CCR1 antagonists may have therapeutic applications in aAMD. Further experiments are required to assess the effect of CCR1 inhibition in rd10 and senescent mice and validate the protective effect of CCR1 antagonist during retinal degeneration. Nevertheless, recent studies performed on human AMD materials revealing an increased expression/secretion of CCR1 and its ligands (*Joo et al., 2021*; *Saddala et al., 2019*), supporting the notion that CCR1 could represent a potential therapeutic target for treating AMD disease.

Our data support the notion that infiltrating monocytes interact with Müller cells during retinal disease, with deleterious effects. Although largely known for their structural role in the retina, Müller cells also play an essential role in maintaining metabolic homeostasis and function in the retina. For example, Müller cells can exchange ions, water, and bicarbonate molecules in order to regulate the composition of the extracellular fluid, and these cells use a variety of complex mechanisms to regulate synaptic activity, guide incoming light, and both support and protect neurons (*Reichenbach and Bringmann, 2013*). Importantly, Müller cells also serve as a source of cytokines and growth factors that drive neuronal and immune responses (*Abcouwer, 2017*; *Coughlin et al., 2017*). During ocular inflammation, Müller cells are activated by a process known as gliosis, which allows them to interact with immune cells and microglial cells recruited to the site of inflammation (*Bringmann et al., 2006*). With respect to their role in pathogenesis, previous studies suggest that Müller cells are associated with the progression of several inflammatory eye diseases such as diabetic retinopathy (*Capozzi et al., 2018*) by activating the CD40 receptor (*Portillo et al., 2014*; *Portillo et al., 2016*; *Portillo et al., 2017*) or by acting upon the microvascular to promote angiogenesis (*Sugiyama et al., 2004*; *Xin et al., 2013*).

Müller cells have also been shown to promote the development of glaucoma in an experimental model of chronic ocular hypertension (*Zhong-feng and Xiong-li, 2016*). In addition, other studies suggest that these cells play a role in the development of proliferative vitreoretinopathy (*Bringmann and Wiedemann, 2012*), retinitis pigmentosa, and AMD (*Massengill et al., 2018*). Indeed, drusen formation has been associated with Müller cells gliosis (*Telegina et al., 2018*; *Wu et al., 2003*). However, the mechanism by which Müller cells contribute to the progression of AMD has not been identified. Here, we show that CCR1 expression in Müller cells is correlated directly with retinal function in an animal model that recapitulates many of the features associated with aAMD. Furthermore, we show that inhibiting CCR1 reduces Müller cell activation and reduces both retinal inflammation and photoreceptor cell loss.

Despite having distinct etiologies, both retinitis pigmentosa and AMD culminate in the loss of photoreceptor cells. Using models for both diseases, we found that CCR1 expression in Müller cells is correlated with photoreceptor cell death. Although previous studies have shown that Müller cells can directly cause the death of retinal ganglion cells (*Xue et al., 2016*) and endothelial cells (*Portillo et al., 2016*), the notion that Müller cells can be activated by neurotoxic macrophages—and thus may directly cause the death of photoreceptor cells—is novel and warrants further study.

Müller cells were shown previously to induce photoreceptor cell death by recruiting immune cells (*Matsumoto et al., 2018*) and through crosstalk with microglial cells (*Wang et al., 2011*). Similarly, we found that inhibiting CCR1 reduced macrophage infiltration and prevented activation of microglial cells. Interestingly, previous studies found that the chemokine receptor ligand CCL2 can act as an inflammatory cytokine, promoting photoreceptor cell death by recruiting macrophages (*Nakazawa et al., 2007*), while other studies found that the ligand CXCL10 can activate microglial cells via the CCR3 receptor (*Clarner et al., 2015*). Here, we found that inhibiting CCR1 was associated with decreased expression of CCL2 and—albeit to a slightly lesser extent—CXCL10 in Müller cells. Taken together, these findings suggest that CCR1 may play an essential role in the activation of Müller cells, thus leading to photoreceptor cell death.

CCR1 was first identified as a chemokine receptor expressed in specific immune cell types such as monocytes (*Tsou et al., 1998*), which are recruited during photic injury (*Rutar et al., 2015*). Although our results provide the first evidence that CCR1 is expressed in Müller cells, we cannot exclude the possibility that the systemically injected CCR1 inhibitor BX471 may have prevented the infiltration of immune cells in the retina, thereby reducing retinal inflammation and protecting photoreceptor cells from apoptosis. In addition, we found that BX471 significantly decreased the number of macrophages recruited to the photic-injured retina in vivo and reduced the capacity of M2a hMdϕs to attract human

monocytes in vitro. Moreover, we found that inhibiting CCR1 altered the functional properties of M2a hMdɸs, rendering them less neurotoxic by reducing their production of ROS.

Reducing the recruitment of monocytes by inhibiting chemokine receptors has been explored as a possible therapeutic strategy for treating several inflammatory diseases (*Sennlaub et al., 2013*). With respect to ocular inflammatory disease, a recent clinical trial tested a dual CCR2/CCR5 antagonist for treating diabetic macular edema; however, the results of the phase 2 clinical trial showed that this treatment was inferior to currently approved treatments (*Gale et al., 2018*). This is consistent with the suggested role of Müller cells in photoreceptor cell death. Although we found that both CCR2 and CCR5 were upregulated in the photic injury model and in rd10 mice, these receptors do not appear to be expressed in Müller cells. Finally, although CCR1 antagonists have been tested in clinical trials for the treatment of endometriosis and leukemia, their therapeutic value with respect to ocular diseases has not been investigated.

In summary, our results indicate that the mechanism underlying CCR1-mediated photoreceptor cell death seems to include an intrinsic retinal process involving the activation of Müller cells, as well as the recruitment of neurotoxic macrophages to the retina and the functional modulation of M2a hMdɸs. These complementary roles played by CCR1—which include gliosis and the recruitment and polarization of macrophages in the retina—suggest that this receptor may serve as a promising new target for treating ocular degenerative diseases such as aAMD.

## Materials and methods

### Patients

A total of 33 patients with AMD (14 women and 19 men) 77.1±3 years of age (range: 63–95 years) were recruited at the Retina Clinic in the Department of Ophthalmology at the Hadassah-Hebrew University Medical Center in Jerusalem, Israel. The criteria for establishing a diagnosis of AMD included >55 years of age and clinical findings of intermediate or advanced AMD in accordance with the 1999 AREDS (Age-Related Eye Disease Study) criteria (*Age-Related Eye Disease Study Research, 1999*). Moreover, we excluded eyes with high myopia (>6 diopters), trauma, other retinal disease, and/or uveitis. We also excluded patients who presented with a major systemic illness such as cancer, autoimmune disease, congestive heart failure, and/or uncontrolled diabetes. All participating patients provided written informed consent, and the study was approved by our institutional ethics committee.

### Preparation of monocytes and macrophages

Whole blood samples (30 ml) were collected from patients with AMD in EDTA tubes (BD Biosciences, Franklin Lakes, NJ, USA). Monocytes were then isolated from these whole blood samples using negative selection as described previously (*Grunin et al., 2012*). In brief, PBMCs were separated from the whole blood using a Histopaque-Ficoll gradient (Sigma-Aldrich, Munich, Germany) and washed twice by centrifugation at 1500 rpm for 10 min to remove the platelets; live cells were counted using a hemocytometer with the trypan blue exclusion method. Total blood monocytes, including $CD14^{++}CD16^{-}$ and $CD14^{+}CD16^{+}$ monocytes, were then isolated using the EasySep negative selection kit (Stemcell Technologies, Vancouver, Canada) in accordance with the manufacturer's instructions.

To prepare macrophages, PBMCs were isolated from the whole blood samples as described above, stimulated with M-CSF (macrophage colony-stimulating factor; PeproTech, Rocky Hill, NJ, USA) to produce non-activated (M0) macrophages, and then activated with either IFN-γ and LPS (to produce M1 hMdɸs), IL-4 and IL-13 (to produce M2a hMdɸs), or IL-10 (to produce M2c macrophages) as previously described (*Bouhlel et al., 2007*; *Gelinas et al., 2011*; *Mantovani et al., 2002*; *Martinez, 2009*; *Pelegrin and Surprenant, 2009*). In brief, PBMCs were suspended in RPMI 1640 medium (Biological Industries, Kibbutz Beit-Haemek, Israel) and seeded at $3\times10^{7}$ cells/cm$^{2}$ in six-well plates. The monocytes were then incubated at 37°C in 5% $CO_2$ for 2 hr, washed with PBS, and then cultured for 7 days in RPMI 1640 supplemented with 10% (vol/vol) fetal calf serum (FCS), 1% non-essential amino acids, 2 mmol/L L-glutamine, 1 mM sodium pyruvate, 100 units/ml penicillin, 100 µg/ml streptomycin, and 50 ng/ml M-CSF; M-CSF was included in the growth medium to drive maturation of the monocytes into macrophages. M1 hMdɸs were obtained by the addition of 20 ng/ml IFN-γ (PeproTech) and 100 ng/ml LPS (Sigma-Aldrich) on day 6, M2a hMdɸs were obtained by the addition of 50 ng/ml IL-13

(PeproTech) and 20 ng/ml IL-4 (PeproTech) on day 5, and M2c hMdɸs were obtained by the addition of 50 ng/ml IL-10 (PeproTech) on day 5. hMdɸ cells that were not activated were classified as unpolarized HMdɸs (M0). M1 macrophages require 24 hr for polarization, whereas M2a and M2c cells require 48 hr (*Allavena et al., 1998*); therefore, the hMdɸs were polarized on different days so that the in vitro and in vivo experiments could be performed on the same day.

## Macrophage co-cultures with mouse retinal explants

The various groups of polarized hMdɸs were harvested and seeded for a minimum of 2 hr on a polycarbonate filter in serum-free DMEM (Biological Industries) supplemented with glutamine and penicillin-streptomycin. In parallel, 6-week-old C57BL/6 mice (n=9) were anesthetized and euthanized via cervical dislocation. Both eyes (n=18) were then enucleated and placed in cold serum-free DMEM supplemented with glutamine and penicillin-streptomycin. The retinas were gently detached from the choroid tissue and immediately placed on the polycarbonate filter so that the hMdɸs were in contact with the photoreceptor layer. For retinal explants cultivated without direct contact with hMdɸs (n=18), $10^5$ hMdɸs were seeded in the bottom well of a Boyden chamber, and the explant was placed in the upper chamber. After incubation for 18 hr, the mouse retinas were fixed in 4% paraformaldehyde (PFA) for 30 min at room temperature (RT) and then permeabilized for 30 min on ice in methanol, followed by 30 min on ice in a 2:1 mixture of methanol/acetone. Terminal deoxynucleotidyl transferase dUTP nick end labeling (TUNEL) staining was then performed using an In Situ Cell Death Detection Kit, TMR red (La Roche, Basel, Switzerland) in accordance with the manufacturer's instructions. For rhodopsin and RPE65 immunostaining, retinal explants were incubated with either mouse anti-rhodopsin (10 µg/ml; ab3267, Abcam, Cambridge, UK) or mouse anti-RPE65 (10 µg/ml; ab13826, Abcam) overnight at 4°C, washed, and then incubated with donkey anti-mouse IgG-Alexa Fluor 488 (1 µg/ml, ab150109, Abcam) for 1 hr at RT. A Zeiss LSM 710 confocal microscope was used to visualize the TUNEL-stained cells in 11 randomly selected retinal fields.

## Reverse transcription and real-time qPCR

RNA was extracted from isolated hMdɸs and retina samples using TRIzol Reagent (Sigma-Aldrich) in accordance with the manufacturer's instructions. The RNA quality and quantity were measured using a NanoDrop spectrophotometer (Thermo Scientific, Waltham, MA, USA) and a bioanalyzer (Agilent

**Table 1.** qPCR Primer Sequences.

| Gene | Forward (5'–3') | Reverse (5'–3') |
| --- | --- | --- |
| Human *GAPDH* | AACAGCCTCAAGATCATCAGC | GGATGATGTTCTGGAGAGCC |
| Human CXCL10 | GGTGAGAAGAGATGTCTGAATCC | GTCCATCCTTGGAAGCACTGCA |
| Human CCL17 | AGGGATGCCATCGTTTTTGTAA | GCTTCAAGACCTCTCAAGGCT |
| Human CD163 | CAGTGCAGAAAACCCCACAA | AAAGGATGACTGACGGGATGA |
| Human *CCL13* | ATCTCCTTGCAGAGGCTGAA | CTTCTCCTTTGGGTCAGCAC |
| Human *CCL23* | TTTGAAACGAACAGCGAGTG | CAGCATTCTCACGCAAACC |
| Human *CCL14* | ATACAGCTAAAGTTGGTGGGGG | TGGTGATGAAGACAATTCCGGG |
| Human *CCR1* | AAGTCCCTTGGAACCAGAGAGAAG | CCAACCAGGCCAATGACAAA |
| Mouse *Gapdh* | AACTTTGGCATTGTGGAAGG | ACACATTGGGGGTAGGAACA |
| Mouse *Ccr1* | GTTGGGACCTTGAACCTTGA | CCCAAAGGCTCTTACAGCAG |
| Mouse *Ccr2* | GAAGAGGGCATTGGATTCAC | TATGCCGTGGATGAACTGAG |
| Mouse *Ccr5* | TCTCCTAGCCAGAGGAGGTG | TGTCATAGCTATAGGTCGGAACTG |
| Mouse *Adgre1** | GCATCATGGCATACCTGTTC | AGTCTGGGAATGGGAGCTAA |
| Mouse *Ccl2* | AGGTCCCTGTCATGCTTCTG | TCTGGACCCATTCCTTCTTG |
| Mouse *Cxcl1* | GACCATGGCTGGGATTCACC | CCAAGGGAGCTTCAGGGTCA |
| Mouse *Cxcl10* | CATCCCTGCGAGCCTATCC | CATCTCTGCTCATCATTCTTTTTCA |

*Encodes the F4/80 protein, also known as EMR1 (EGF-like module-containing mucin-like hormone receptor-like 1).

Technologies, Santa Clara, CA, USA), and RNA was reverse-transcribed to create cDNA using the qScript cDNA Synthesis Kit (Quantabio, Beverly, MA, USA) in accordance with the manufacturer's instructions. qPCR was then performed using the PerfeCTa SYBR Green FastMix kit (Quantabio); the gene-specific primers (Sigma-Aldrich) used in this study are listed in *Table 1*. Each gene was amplified in triplicate, and the expression level of each gene was normalized to human *GAPDH* or mouse *Gapdh* as an endogenous control using the standard 2(ΔΔCT) method (*Livak and Schmittgen, 2001*).

## Photo-oxidative retinal injury and intravitreal injections

Albino BALB/c mice that were homozygous for the wild-type *Crb1*, *Gnat2*, and *Rpe65* genes were used for this study (*Chang et al., 2013*; *Mattapallil et al., 2012*; *Wenzel et al., 2003*). Photic injury was induced essentially as described previously (*Grimm and Remé, 2013*), with optimization to ensure an approximately 50% reduction in ONL thickness as previously described (*Elbaz-Hayoun et al., 2019*). In brief, after 1 hr of dark adaptation, 6-week-old BALB/c mice raised under a standard light/dark cycle were exposed to 8000 lux of white light for 3 hr. Photic injury was then induced as follows, according to the appropriate circadian rhythm: the pupils were dilated with Cyclogyl (one drop per eye, Sandoz Farmaceutica S.A., Madrid, Spain) and 5% phenylephrine (Fischer Pharmaceutical Labs, Tel Aviv, Israel) at 9:30 am under a red light; the light level was adjusted at 9:45 am, and photic injury was induced for 3 hr (from 10:00 am to 1:00 pm), during which the mice were placed in a cage (maximum two mice per cage) lined with aluminum foil, and the temperature was maintained below 30°C.

Immediately after photic injury, the mice received an intravitreal injection of either human monocytes or hMdɸs that were labeled with the Vybrant DiO tracer (Invitrogen-Molecular Probes, Carlsbad, CA, USA), and then returned to the standard light/dark cycle. For these experiments, we chose the intravitreal route over the subretinal route in order to avoid triggering an immune response due to RPE immunogenicity and the potentially higher risk of RPE and/or Bruch's membrane breakthrough (*Westenskow et al., 2015*); moreover, intravitreal injection allows the injected cells to distribute across the entire retina, creating a wider and less-biased effect on ONL thickness. To compare the effect of monocytes from AMD patients with those from unaffected aged-matched control, $10^5$ human monocytes extracted from AMD patients were injected into the right eye, while the left eye received an injection of monocyte from an unaffected individual.

To assess monocyte or hMdɸs from AMD patients, for each mouse, $10^5$ human monocytes or hMdɸs suspended in PBS were injected into the right eye, while the left eye received an injection of PBS as a control. As an additional control, some mice were not exposed to light and did not receive an intravitreal injection of monocytes or hMdɸs. An antibiotic ointment (5% chloramphenicol) was applied after each intravitreal injection.

## CCR1 blocking treatment

Immediately after photic injury was induced, the mice received subcutaneous injections of either the CCR1-specific antagonist BX471 (50 mg/kg body weight; Tocris, Bristol, UK) or vehicle (40% cyclodextrin in saline) every 12 hr for 5 days. For injection, BX471 was dissolved at a final concentration of 10 mg/ml in saline containing 40% (wt/vol) cyclodextrin (Sigma-Aldrich); the solution was mixed thoroughly and dissolved overnight at 4°C, after which the pH was adjusted to 4.5 with NaOH, and the solution was filtered through a 0.45 µm filter.

## Electroretinography recording and in vivo retinal imaging

Seven days after photic injury, the pupils were dilated with tropicamide (Fischer Pharmaceutical Labs) and phenylephrine (Fischer Pharmaceutical Labs), and the corneas were kept moist by application of carboxymethylcellulose (Fischer Pharmaceutical Labs). Retinal images were obtained using a Spectralis Optical Coherence Tomography device and a Micron III retinal microscope (Phoenix Research Labs, San Francisco, CA, USA). Blue autofluorescence images were obtained using an excitation wavelength of 488 nm, and full-field ERG was performed in dark-adapted mice. During ERG recording, the eyes were anesthetized with oxybuprocaine hydrochloride drops (Fischer Pharmaceutical Labs). All procedures were performed in dim red lighting or in total darkness, and the mice were kept warm throughout the recording. During the recording, the mouse was positioned facing the center of a Ganzfeld bowl, ensuring equal, simultaneous illumination of both eyes. ERG data were recorded inside a Faraday cage using an Espion computerized system (Diagnosys LLC, Littleton, MA, USA).

Dark-adapted ERG responses to a series of white flashes at increasing intensity (from 0.000006 to 9.6 cd·s/m$^2$) were recorded at inter-stimulus intervals increasing from 10 s (for the lowest-intensity flashes) to 90 s (for the highest-intensity flashes). Light adaptation was performed using a background illumination of 30 cd/m$^2$. For analysis, the b-wave amplitude was measured from the trough of the a-wave to the peak of the b-wave.

## Immunohistochemistry

Seven days after photic injury, the mice were euthanized, the eyes were enucleated and sectioned at 10 µm using a cryostat, and the sections were immunostained as previously described (*Hagbi-Levi et al., 2017*). In brief, the eyes were fixed in 4% PFA for 2 hr and then placed in 30% sucrose overnight at 4°C. The eyes were then placed in optimal cutting temperature compound (Scigen Scientific, Gardena CA, USA), and 10 µm sections were cut and placed in blocking solution (PBS containing 10% serum and 0.1% Triton X-100) for 1 hr at RT. The sections were then incubated in primary antibody overnight at 4°C; the following day, the sections were incubated in secondary antibody for 1 hr at RT. The nuclei were counterstained with DAPI, and the sections were visualized using a fluorescence microscope. For hMdɸ labeling, after polarization the cells were washed once with PBS and then fixed with 4% PFA for 30 min at RT; after three washes with PBS, the cells were incubated in blocking solution and immunostained with primary and secondary antibodies as described above.

The following primary antibodies were used for these experiments: rabbit anti-4 HNE antibody (10 µg/ml; ab46545, Abcam), rat anti-cd11b antibody (0.5 µg/ml; ab64347, Abcam), mouse anti-human CCR1 (25 µg/ml; mab145, R&D Systems, Minneapolis, MN, USA), rat anti-mouse CCR1 (25 µg/ml; mab5986, R&D Systems), rat anti-mouse-CCR5 (10 µg/ml; ab11466, Abcam), rabbit anti-mouse CCR2 (5 µg/ml; NBP2-67700, Novus Biologicals, Littleton, CO, USA), rabbit anti-mouse GFAP (0.5 µg/ml; ab64347, Abcam), and rabbit anti-mouse Iba1 (5 µg/ml; ab153696, Abcam). The following secondary antibodies were used: donkey anti-mouse IgG-Alexa Fluor 488 (ab150109; Abcam), donkey anti-rat IgG-Alexa Fluor 555 (ab150154; Abcam), and goat anti-rabbit IgG-Alexa Fluor 488 (ab150085; Abcam). TUNEL staining was performed using the In Situ Cell Death Detection Kit, TMR red (La Roche, Basel, Switzerland) in accordance with the manufacturer's instructions.

To measure the thickness of the ONL, the sections were stained with DAPI and the number of photoreceptor nuclei was counted at fixed distances from the ONH.

## Measurement of ROS

Human hMdɸs were cultured in six-well plates for 6 days and polarized as described above. To block CCR1, 0.5 µM or 5 µM BX471 was added to M2a hMdɸ cultures for 1 hr at 37°C. ROS production was measured using the DCFDA Cellular ROS Detection Assay (ab113851; Abcam) and a fluorescence microplate reader (Tecan Group, Männedorf, Switzerland) in accordance with the manufacturer's instructions.

## Multiplex ELISA

After the macrophages were polarized (see above), the culture medium was collected and stored at −80°C. A panel of 120 cytokines was then measured in the culture medium using the Human Cytokine Array GS2000 (RayBiotech Life, Inc, Norcross, GA, USA) in accordance with the manufacturer's instructions (*Table 2*).

## In vitro migration assay

CD14$^{++}$CD16$^-$ and CD14$^+$CD16$^+$ monocytes were isolated as described above, and their migration was measured using a 24-well Boyden chamber assay (Corning, 5 µm pore size). In brief, conditioned medium obtained from 10$^5$ M1 hMdɸs or 10$^5$ M2a hMdɸs was harvested, centrifuged to remove cell debris, and placed in the bottom chamber of the plate. Next, 1.2×10$^5$ previously isolated monocytes suspended in 100 µl RPMI with 1% FCS were placed in the upper chamber.

To determine the effect of blocking CCR1 on monocyte migration, 10$^5$ M2a hMdɸs were first grown in the bottom chamber for 48 hr. The day of the experiment, fresh RPMI containing 1% FCS was used to fill the bottom chamber, and 1.2×10$^5$ previously isolated monocytes that were pretreated for 1 hr with either 10 µM BX471 or vehicle were placed in the upper chamber containing 100 µl RPMI with

**Table 2.** Multiplex ELISA results.

| p-Value | M1:M2a ratio | M2a | M1 | Protein name |
|---------|-------------|-----|-----|--------------|
| 0.0123 | 0.00 | 373 | 0 | TGFa |
| 0.0009 | 0.10 | 7457 | 826 | CCL14 |
| 0.0001 | 0.11 | 36,571 | 3821 | CCL13 |
| 0.0020 | 0.27 | 14,645 | 3866 | CCL17 |
| 0.0097 | 0.51 | 6530 | 3007 | CCL23 |
| 0.0159 | 0.52 | 82,038 | 43,946 | CCL24 |
| 0.0409 | 0.70 | 74,990 | 48,575 | PDGFB |
| 0.0298 | 0.78 | 937 | 712 | IL-7 |
| 0.0287 | 0.81 | 869 | 688 | TNFb |
| 0.0026 | 99.24 | 1458 | 109,591 | CXCL13 |
| 0.0558 | 40.72 | 394 | 9887 | CSF3 |
| 0.00002 | 39.66 | 16,545 | 238,286 | IL-6 |
| 0.0002 | 17.91 | 635 | 7254 | CXCL11 |
| 0.0002 | 15.27 | 2979 | 32,877 | CCL19 |
| 0.0008 | 6.33 | 5322 | 30,523 | CCL20 |
| 0.0032 | 5.47 | 28,468 | 94,382 | CCL5 |
| 0.0108 | 4.74 | 952 | 4515 | VEGF |
| 0.0210 | 3.08 | 5085 | 14,169 | IL-10 |
| 0.0572 | 2.60 | 6683 | 18,212 | CXCL6 |
| 0.0219 | 2.50 | 14,774 | 27,516 | CCL1 |
| 0.0029 | 2.43 | 150 | 296 | CXCL12 |
| 0.0008 | 2.27 | 8696 | 19,269 | CXCL10 |
| 0.0019 | 1.40 | 31,868 | 44,135 | CCL7 |
| 0.0506 | 1.21 | 365 | 442 | IL-11 |

1% FCS. After incubation for 2 hr at 37°C, the monocytes that migrated to the bottom chamber were measured using FACS analysis.

## Flow cytometry

To evaluate the effect of polarized hMdɸs on photoreceptor apoptosis in vitro, retinal explants that were either in contact with or not in contact with polarized hMdɸs were fixed, TUNEL stained as described above, and digested using a homogenizer in 500 ml PBS containing 100 mg/ml collagenase/dispase (10-269-638; La Roche). Samples (100 ml each) were then placed in tubes, washed twice with 2 ml FACS washing buffer containing 0.5% (wt/vol) BSA in PBS, and centrifuged at 1500 × *g* for 5 min to collect the cells. The cells were then filtered through a 60 µm mesh, and fluorescence intensity was immediately read using an LSR-II flow cytometer (BD Biosciences, Franklin Lakes, NJ, USA) in accordance with the manufacturer's instructions.

To measure the level of CCR1 expression in M1 and M2a hMdɸs, the cells were fixed in 4% PFA for 20 min at RT and washed twice with PBS. The cells were then stained with mouse anti-human CCR1 antibody (2.5 µg/10⁶ cells; mab145, R&D Systems) for 20 min at RT, followed by donkey anti-mouse IgG-Alexa Fluor 488 antibody (1 µg/ml; ab150109, Abcam) for 20 min in the dark at RT. Each sample was then washed twice with PBS containing 0.5% (wt/vol) BSA and centrifuged at 1500 rpm for 5 min to collect the cells. The cells were then filtered through a 60 µm mesh, and fluorescence intensity was

immediately read using an LSR-II flow cytometer (BD Biosciences) in accordance with the manufacturer's instructions.

## Statistical analysis

The appropriate statistical tests were used based on the results of a test for normalcy and the sample distribution and parameters. The biostatistical software package InStat (GraphPad Software, San Diego, CA, USA) was used for data analysis. Data were analyzed using a one-way ANOVA followed by the Tukey-Kramer post hoc test or an unpaired Student's t-test, where appropriate. Details of the number of replicates and the specific statistical test used are provided in the individual figure legends. Values over 2 standard deviations from the average were excluded from statistical analysis. The results are presented as the mean fold change ± the standard error of the mean (SEM).

## Additional information

### Funding

| Funder | Grant reference number | Author |
|---|---|---|
| Israel Science Foundation | 1006/13 | Sarah Elbaz-Hayoun |
| Israel Science Foundation | 3485/19 | Sarah Elbaz-Hayoun |
| Israeli Ministry of Health | 9184 | Sarah Elbaz-Hayoun |

The funders had no role in study design, data collection and interpretation, or the decision to submit the work for publication.

### Author contributions

Sarah Elbaz-Hayoun, Conceptualization, Investigation, Writing – original draft; Batya Rinsky, Validation; Shira Hagbi-Levi, Methodology; Michelle Grunin, Formal analysis; Itay Chowers, Conceptualization, Supervision

### Author ORCIDs

Sarah Elbaz-Hayoun ⓘ http://orcid.org/0009-0006-8605-5775
Shira Hagbi-Levi ⓘ http://orcid.org/0000-0002-2891-0079
Michelle Grunin ⓘ http://orcid.org/0000-0002-3155-2858
Itay Chowers ⓘ http://orcid.org/0000-0003-1897-4973

### Ethics

Ethical approval for all experimental protocols and studies involving human subjects were provided by the Local Committee on Research Involving Human Subjects of the Hebrew University-Hadassah Medical School, the Helsinki Committee of Hadassah Medical Organization, and the Israel Ministry of Health's Helsinki Committee for Genetic Experiments on Human Subjects (File # 22-03.08.07). All patients and controls signed informed consent forms that adhered to the tenets of the Declaration of Helsinki before participating in the study.

Ethical approval for all protocols involving animals was approved by the Authority for Biological and Biomedical Models (ABBM) and the University Ethics Committee for the Care and Use of Laboratory Animals at the Hebrew University, which is certified by the Association for Assessment and Accreditation of Laboratory Animal Care (AAALAC) (ethical approval number: MD-16-14796-3, NIH approval number: OPRR-A01-5011). All researchers working with laboratory animals received the approval of the ethics committee of the ABBM to ethically work with laboratory animals. All guidelines with regard to the humane and ethical treatment of laboratory animals (from ARVO) were followed to the utmost and all the methods used in this study were carried out in accordance with the approved guidelines for the study.

### Decision letter and Author response

Decision letter https://doi.org/10.7554/eLife.81208.sa1
Author response https://doi.org/10.7554/eLife.81208.sa2

## Additional files

### Supplementary files
- MDAR checklist
- Supplementary file 1. Cytokines array result.

### Data availability
All data generated or analysed during this study are included in the manuscript and supporting file.

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
