## [Editor Report]

Immune cell invasion, gliosis, and photoreceptor cell death are observed in multiple retinal diseases. This important study identifies cells and signaling pathways that connect these three processes during retinal degeneration. The authors provide convincing experimental evidence linking macrophages to the activation of retinal Müller glial cells and photoreceptor death. These results are significant as they identify cell types and potential targets linking immune cells to retinal cell changes, ultimately resulting in photoreceptor cell death.

---

## [Decision Letter]

**Decision letter after peer review:**

Thank you for submitting your article "CCR1 Mediates Müller Cell Activation and Photoreceptor Cell Death in Macular and Retinal Degeneration" for consideration by *eLife*. Your article has been reviewed by 3 peer reviewers, and the evaluation has been overseen by a Reviewing Editor and Mone Zaidi as the Senior Editor. The following individuals involved in the review of your submission have agreed to reveal their identity: Michael E Zuber (Reviewer #1); Rajendra S. Apte (Reviewer #2).

Essential revisions:

Main controls to include for a revision would be injecting non-AMD monocytes as controls, tracing monocyte development into macrophages, proof that macrophages migrated into the subretinal space, appropriate M1 markers, a demonstration that M2a cells induce apoptosis in vivo, improving ROS-related data, and more on MPIF1 or MCP4.

Discussion of other main concerns would be that the photic injury model is limited in its reflection of human AMD. Also, the rd10 model of phosphodiesterase is a model of retinitis pigmentosa and not AMD.

*Reviewer #2 (Recommendations for the authors):*

It would be interesting to examine the role of these M2a cells that are identified in surrogate murine models of retinal disease e.g. laser injury-induced CNV or lipid-induced retinal degeneration

The role of macrophage polarization was examined and published in retinal models starting in 2006. As the authors develop their interesting research based on these initial studies it would be appropriate to reference the papers listed below so that the literature is presented in the appropriate context:

PMID: 17975672

PMID: 16903779

PMID: 31422901

PMID: 23562078

*Reviewer #3 (Recommendations for the authors):*

The authors perform an elegant study where they show that intravitreal injection of human monocytes from patients with AMD cause reduced ERG B-wave amplitudes and photoceptor cell loss compared to controls in the photic retinal injury model. Differentiation of human monocytes from patients with AMD into M2a macrophages caused increased photoreceptor cell loss compared to M1 macrophages. This is an interesting finding as I may have expected that pro-inflammatory M1 macrophages caused more photoreceptor loss. Could the authors clarify how many times the experiment was repeated? I see that n=8 mice were used for each group; however, it's important that the experiment be repeated multiple times. Next, the authors show that after co-culturing retinal explants with M1 and M2a human macrophages followed by TUNEL staining, M2 human macrophages had significantly more apoptotic photoreceptor cells than M1 human macrophages. The authors show that human M2a macrophages have significantly more ROS compared to M0 and M1 human macrophages; however, injection of human M2a macrophages did not cause increased oxidative damage compared to control conditions. Using a multiplex cytokine assay of 120 cytokines between human M1 and M2a macrophages-conditioned medium, the authors found increased levels of 9 cytokines, including three HCC-1, MCP-4, and MPIF-1, which are ligands of the C-C chemokine receptor CCR1. Co-staining showed CCR1 expression in Muller cells following photic injury. In the rd10 mouse model of retinal degeneration as well as aged BALB/c mice CCR1 is upregulated in Muller cells. Injection of mice with the CCR1-specific inhibitor BX471 caused increased photoreceptor numbers and B-wave amplitudes in the photic-injury model. Overall the experiments are well performed; however, the photic injury model is limited in its reflection of human AMD. While it's outside of the scope of this paper, it would be nice to see these pathways validated in human-diseased AMD tissue. I agree with the authors that human M2a macrophages mediate neurotoxicity in the photic injury model; however, the limitation of this study is that the authors don't directly provide evidence of this pathway in human tissue and may not accurately represent human AMD. Further, the rd10 model of phosphodiesterase is a model of retinitis pigmentosa and not AMD, so it's possible that the CCR1 inhibitor may be more applicable in retinitis pigmentosa than AMD. It's also unclear whether the role of CCR1 activation is mediated through Muller cells or immune cell infiltration as monocytes express CCR1. Thus additional research would need to be performed to define the cell-type specificity of CCR1-inhibition in this model and to show whether CCR1 antagonists are translational to human AMD. However, this is a well-performed study and of interest to the field.

---

## [Author Response]

Essential revisions:Main controls to include for a revision would be injecting non-AMD monocytes as controls, tracing monocyte development into macrophages, proof that macrophages migrated into the subretinal space, appropriate M1 markers, a demonstration that M2a cells induce apoptosis in vivo, improving ROS-related data, and more on MPIF1 or MCP4.Discussion of other main concerns would be that the photic injury model is limited in its reflection of human AMD. Also, the rd10 model of phosphodiesterase is a model of retinitis pigmentosa and not AMD.Reviewer #2 (Recommendations for the authors):It would be interesting to examine the role of these M2a cells that are identified in surrogate murine models of retinal disease e.g. laser injury-induced CNV or lipid-induced retinal degeneration

Thank you for this important comment. We previously assess the role of M1 and M2a cells from AMD patients in laser injury- induced CNV in rat (Hagbi-Levi S et al., 2016). We showed that adoptive transfer of M1 cells is associated with larger CNV compared with adoptive transfer of M2a macrophages.

The role of macrophage polarization was examined and published in retinal models starting in 2006. As the authors develop their interesting research based on these initial studies it would be appropriate to reference the papers listed below so that the literature is presented in the appropriate context:PMID: 17975672PMID: 16903779PMID: 31422901PMID: 23562078

Thank you for this comment. We have added references to these important earlier reports that provided solid information and the basis to continue research in the field.

PMID: 17975672- Line 99

PMID: 16903779- Line 78

PMID: 31422901- Line 86

PMID: 23562078- Line 99

Reviewer #3 (Recommendations for the authors):The authors perform an elegant study where they show that intravitreal injection of human monocytes from patients with AMD cause reduced ERG B-wave amplitudes and photoceptor cell loss compared to controls in the photic retinal injury model. Differentiation of human monocytes from patients with AMD into M2a macrophages caused increased photoreceptor cell loss compared to M1 macrophages. This is an interesting finding as I may have expected that pro-inflammatory M1 macrophages caused more photoreceptor loss. Could the authors clarify how many times the experiment was repeated? I see that n=8 mice were used for each group; however, it's important that the experiment be repeated multiple times.

The in vivo experiments were repeated 8 times; each of the blood samples was from a different AMD patient, and PBMCs were isolated and polarized into M0, M1, M2a and M2c macrophages from each patient. Additionally, we have validated the results in ex-vivo experiments with samples from different patients, testing 6 replicates, and the results were in agreement with the in vivo experiments. It seems that polarized macrophages can have variable effect depending on the microenvironmental context and the origin of the macrophages. For example, we have previously demonstrated that polarized macrophages from AMD patients and unaffected controls can have different effects (Hagbi-Levi et al. 2016).

Next, the authors show that after co-culturing retinal explants with M1 and M2a human macrophages followed by TUNEL staining, M2 human macrophages had significantly more apoptotic photoreceptor cells than M1 human macrophages. The authors show that human M2a macrophages have significantly more ROS compared to M0 and M1 human macrophages; however, injection of human M2a macrophages did not cause increased oxidative damage compared to control conditions. Using a multiplex cytokine assay of 120 cytokines between human M1 and M2a macrophages-conditioned medium, the authors found increased levels of 9 cytokines, including three HCC-1, MCP-4, and MPIF-1, which are ligands of the C-C chemokine receptor CCR1. Co-staining showed CCR1 expression in Muller cells following photic injury. In the rd10 mouse model of retinal degeneration as well as aged BALB/c mice CCR1 is upregulated in Muller cells. Injection of mice with the CCR1-specific inhibitor BX471 caused increased photoreceptor numbers and B-wave amplitudes in the photic-injury model. Overall the experiments are well performed; however, the photic injury model is limited in its reflection of human AMD. While it's outside of the scope of this paper, it would be nice to see these pathways validated in human-diseased AMD tissue. I agree with the authors that human M2a macrophages mediate neurotoxicity in the photic injury model; however, the limitation of this study is that the authors don't directly provide evidence of this pathway in human tissue and may not accurately represent human AMD.

Thank you for this comment. As the reviewer mentioned, the study of human AMD tissue is beyond the scope of the current ms. We have tested macrophages derived from humans and show their function in ex vivo and in vivo models. Thus, results of this study suggest that M2a macrophages may potentially have a role in AMD, but, validation of these findings in humans is required. Nevertheless, recent studies performed in human AMD samples supported the results presented in our manuscript. Joo et al. (2021) showed increased level of two ligands of CCR1, MIP-1α and MIP-1β, in aqueous humour from AMD patients compared with age-matched control. Also, Saddala et al. (2019), revealed increased expression of CCR1, of its ligands (MIP-1α and MIP-1β) and Muller gliosis-related genes (CXCL10, CCL2) in the human peripheral retina and RPE-choroid-sclera of AMD patient. These data support our finding regarding a potential involvement of CCR1 in AMD. The text of the revised ms. was revised accordingly in the Discussion section and the above mentioned references were added to the ms. (Lines 459-464).

Further, the rd10 model of phosphodiesterase is a model of retinitis pigmentosa and not AMD, so it's possible that the CCR1 inhibitor may be more applicable in retinitis pigmentosa than AMD. It's also unclear whether the role of CCR1 activation is mediated through Muller cells or immune cell infiltration as monocytes express CCR1. Thus additional research would need to be performed to define the cell-type specificity of CCR1-inhibition in this model and to show whether CCR1 antagonists are translational to human AMD.

Thank you for this comment. We agree that CCR1 expression on both Muller cells and macrophages can potentially mediate retinal cell death in aAMD. This possibility is discussed in the revised manuscript (Lines 510-519 and 536-542). Our results show that it is possible that CCR1 can potentially mediate retinal cell death in retinitis pigmentosa. We have detailed this issue in the Discussion section of the revised ms. (Lines 459-464).

However, this is a well-performed study and of interest to the field.

Thank you!